# SPARSE MeZO: LESS PARAMETERS FOR BETTER PERFORMANCE IN ZEROTH-ORDER LLM FINE-TUNING

## ABSTRACT

While fine-tuning large language models (LLMs) for specific tasks often yields impressive results, it comes at the cost of memory inefficiency due to back-propagation in gradient-based training. Memory-efficient Zeroth-order (MeZO) optimizers, recently proposed to address this issue, only require forward passes during training, making them more memory-friendly. However, compared with exact gradients, ZO-based gradients usually exhibit an estimation error, which can significantly hurt the optimization process, leading to slower convergence and suboptimal solutions. In addition, we find that the estimation error will hurt more when adding to large weights instead of small weights. Based on this observation, this paper introduces Sparse MeZO, a novel memory-efficient zeroth-order optimization approach that applies ZO only to a carefully chosen subset of parameters. We propose a simple yet effective parameter selection scheme that yields significant performance gains with Sparse-MeZO. Additionally, we develop a memory-optimized implementation for sparse masking, ensuring the algorithm requires only inference-level memory consumption, allowing Sparse-MeZO to fine-tune LLaMA-30b on a single A100 GPU. Experimental results illustrate that Sparse-MeZO consistently improves both performance and convergence speed over MeZO without any overhead. For example, it achieves a 9% absolute accuracy improvement and 3.5x speedup over MeZO on the RTE task.

## 1 INTRODUCTION

Fine-tuning large language models for specific tasks or datasets has become a prevalent practice in machine learning. However, a major obstacle in fine-tuning is the substantial memory requirements, which escalate as models increase in size and complexity, thereby limiting the scalability and accessibility for those with limited computational resources.

To mitigate the memory constraints, Parameter Efficient Fine-Tuning (PEFT) has been developed, allowing for the modification of only a subset of parameters and achieving comparable results to full model tuning (Hu et al., 2021; Lester et al., 2021; Li & Liang, 2021; Zaken et al., 2021; Zhang et al., 2023). However, PEFT methods still necessitate the calculation of gradients for backpropagation and caching of numerous activations during training, which introduces additional memory overhead. For instance, Malladi et al. demonstrates that, even with PEFT, training still requires approximately 6 times more memory than the memory cost for inference. This discrepancy raises a critical question: Can large language models be fine-tuned solely with the cost of inference?

In response to these challenges, zeroth-order (ZO) optimization presents a promising solution (Spall, 1992). ZO optimization is a gradient-free method that estimates gradients using only the forward pass of the model, eliminating the need for backpropagation and, consequently, reducing memory usage. MeZO (Malladi et al., 2023) is a recently proposed zeroth-order method for fine-tuning LLMs that has demonstrated impressive performance. However, compared to exact gradients, ZO-based gradients usually exhibit an estimation error, which can be defined as noise. This noise can significantly hurt the optimization process, leading to slower convergence and suboptimal solutions. Moreover, we find that the estimated ZO gradient is difficult to generalize across batches. Specifically, while it can successfully reduce the training loss on the sampled batch with a high probability, it is more likely to increase the loss on other batches.

To address this challenge, we investigate the impact of gradient noise in zeroth-order optimization for LLM fine-tuning. We measure how the noise affects optimization by evaluating its effect on generalization performance across different data batches. Interestingly, our experiments reveal that the noise has a more significant impact when added to large weights compared to small weights. Based on this finding, we propose a novel sparse memory efficient zeroth-order method (Sparse-MeZO) to selectively optimize small weights, which are more resilient to noise perturbation. By focusing on these noise-resistant weights, we demonstrate that our method enables the use of larger learning rates, leading to improved performance and faster convergence. Our contributions can be summarized as follows:

- In this paper, we investigate the impact of gradient noise in zeroth-order optimization for LLM fine-tuning. Our evaluations show that the gradient noise can make the estimated ZO gradient difficult to generalize across batches and the noise will hurt more when adding to large weights instead of small weights.

- Based on the above finding, we propose a sparse Memory-Efficient Zeroth-Order optimization method Sparse-MeZO (S-MeZO) for large language model fine-tuning. We also provide theoretical analysis to show the convergence of Sparse-MeZO.

- Different from the efficient implementation with random seed in MeZO, we propose a novel memory-efficient implementation of Sparse-MeZO, which can compute the sparse mask and perturb parameters in the forward pass. The technique enables fine-tuning LLaMA-30b with Sparse-MeZO on a single A100 GPU.

- We conduct empirical studies on LLaMA, OPT, and Mistral. The experimental results demonstrate that Sparse-MeZO can improve the fine-tuning performance and yield a faster convergence rate compared with vanilla MeZO across a wide range of natural language processing tasks. For example, it achieves a 9% absolute accuracy improvement and 3.5x speedup over MeZO on the RTE task, as shown in Figure 1.

## 2 PRELIMINARIES

### 2.1 PARAMETER-EFFICIENT FINE-TUNING

Parameter-Efficient Fine-Tuning (PEFT) is designed to facilitate efficient adaptation by updating only a subset of the model's parameters, rather than fine-tuning the entire model (Hu et al., 2021; Zaken et al., 2021). These PEFT approaches can be categorized in various ways. We mainly focus on the selective methods and additive methods.

**Selective Methods.** Selective Methods try to selectively fine-tune a portion of a model and these methods have been explored in various studies. For example, Zaken et al.; Cai et al. focused on the model's bias terms, finding that fine-tuning these terms alone could rival the results of fine-tuning the entire model. However, the effectiveness of this approach diminishes with larger datasets, as shown in further analysis by Zaken et al.. Beyond static parameter adjustments, there has been an exploration into dynamically modifying parts of the model (Brock et al., 2017). This concept was later applied to language models, with AutoFreeze (Liu et al., 2021b) confirming its viability. Nevertheless, these techniques still demand considerable computational resources and sometimes yield less optimal final outcomes.

**Additive Methods.** Additive methods, as an alternative to updating existing parameters, involve incorporating new layers into models, with the fine-tuning process focusing solely on these added layers (Houlsby et al., 2019; Hu et al., 2021; Lin et al., 2020; Rebuffi et al., 2017). Traditional techniques in this category, such as adapters (Houlsby et al., 2019), implemented layer additions in a sequential manner, which unfortunately led to increased inference latency. LoRA (Hu et al., 2021) has been proposed to mitigate this issue, which freezes the weights of the pre-trained model and introduces trainable matrices based on rank decomposition into each layer. Then, it can directly integrate the newly learned weights into the main model. Following this, IA3 (Liu et al., 2022) introduced innovative methods for adding parameters, balancing parameter count with accuracy, while LST (Sung et al., 2022) introduced a highway structure that learns only small, auxiliary channels, aiming to decrease memory demands. Despite these advancements, additive methods generally require meticulous design, and many fail to reduce the computational load during the backward pass.

## 2.2 ZEROTH-ORDER OPTIMIZATION

Unlike traditional gradient-based optimization methods that rely on derivatives to guide the search for optimal solutions, Zeroth-Order (ZO) optimization techniques do not require derivatives for optimization Spall (1992); Liu et al. (2018; 2019). These methods utilize only the value of the objective function, denoted as $f(x)$, at any chosen point $x$. To estimate the gradient in the direction of vector $z$, the objective function is assessed at two points in close proximity, $f(x + \epsilon z)$ and $f(x - \epsilon z)$, with $\epsilon$ being a minimal value. Following this, conventional optimization algorithms, such as gradient descent or coordinate descent, are implemented using these approximated gradient values. Currently, ZO methods have been widely used in various applications, such as adversarial attack and defense (Chen et al., 2017; Ilyas et al., 2018; Tu et al., 2019; Ye et al., 2018), Auto-ML (Ruan et al., 2019; Wang et al., 2022), natural language processing (Sun et al., 2022a;b), reinforcement learning (Vemula et al., 2019), Signal Processing (Liu et al., 2020), and on-chip training (Gu et al., 2021).

### 2.2.1 MEZO

ZO-SGD employs SPSA (Spall, 1992) to estimate the gradient. In general, conventional ZO-SGD algorithms utilizing SPSA consume twice the inference memory. MeZO (Malladi et al., 2023) is a memory-efficient variant of ZO-SGD. It circumvents the storage of gradients by saving the random seed and resampling the same random noise $z$ with the seed during forward process. More specifically, to calculate $\mathcal{L}(\theta + \epsilon z) - \mathcal{L}(\theta - \epsilon z)$, MeZO will sample a noise $z$ to perturb $\theta$ to $\theta + \epsilon z$ and then calculate $\mathcal{L}(\theta + \epsilon z)$. Then it resamples the same noise $z$ with the same seed and move the parameter back $\theta - \epsilon z$ and calculates the loss. As a result, the zeroth order gradient estimator can be computed without any memory overhead.

Figure 1: Performance of MeZO and Sparse-MeZO (S-MeZO) on RTE task. S-MeZO can achieve 3.5x speedup compared with MeZO.

### 2.2.2 SPARSITY FOR ZEROTH-ORDER OPTIMIZATION

The hypothesis proposed by Frankle & Carbin, known as the lottery ticket hypothesis, showed that within a densely connected neural network that is randomly initialized, there exists a subnetwork of sparse yet high-quality connections. Based on the hypothesis, model pruning aims to identify and preserve the crucial 'winning tickets' - sparse subnetworks within the larger neural network that can achieve comparable or even superior performance (Sun et al., 2023; Frantar & Alistarh, 2023). In addition, Dynamic Sparse Training (DST) has been proposed to reduce the training and inference cost in first-order optimization (Liu et al., 2021a; Evci et al., 2020). Recently, several related works have tried to apply the sparsity to zeroth-order optimization (Balasubramanian & Ghadimi, 2018; Cai et al., 2021; 2022; Chen et al., 2023; Gu et al., 2021; Ohta et al., 2020; Wang et al., 2018). For example, DeepZero (Chen et al., 2023) proposes a novel ZO training protocol with model pruning guided sparsity. However, these methods mainly focus on the neural network training from scratch with random initialization, while the application of sparse zeroth-order optimization in fine-tuning tasks remains an area of ongoing exploration.

## 3 PROPOSED METHOD

### 3.1 EMPIRICAL OBSERVATION ON MEZO

For large language models, zeroth-order optimization algorithms like MeZO are often necessary when exact gradients are unavailable or prohibitively expensive to compute. However, compared with exact gradients, these methods inherently introduce noise in the gradient estimates used for optimization.

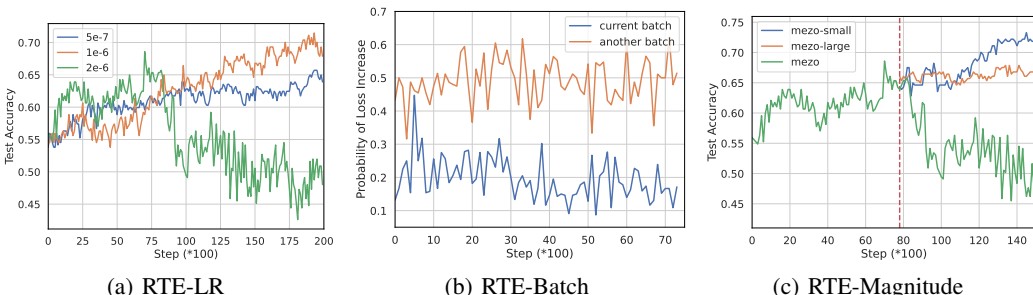

(a) RTE-LR             (b) RTE-Batch             (c) RTE-Magnitude

Figure 2: (a) Test Accuracy with Different Learning Rates on RTE Task. We find MeZO is very sensitive to the selection of learning rate. Even a small increase from $1 \times 10^{-6}$ to $2 \times 10^{-6}$ causes divergence and instability. (b) Probability of Loss Increase on Different Batch. We find the estimated ZO gradient can successfully reduce the loss on the same batch but may be difficult to decrease the loss on the new held-out batch. (c) Continuing training from the drop point with small and large weights. We find that optimizing only the small weights can recover and further improve test accuracy.

Specifically, the zeroth-order gradient $g_z(\theta)$ is approximated as $g_z(\theta) = \frac{\mathcal{L}(\theta+\epsilon z) - \mathcal{L}(\theta-\epsilon z)}{2\epsilon} z$, where $\mathcal{L}$ is the loss function. As shown in Figure 2(a), MeZO exhibits extreme sensitivity to the choice of learning rate. Even a small increase from $1 \times 10^{-6}$ to $2 \times 10^{-6}$ causes divergence and instability, while this larger learning rate is totally fine when fintuning with first-order methods. This suggests that the gradient noise introduced by the zeroth-order approximation, defined as $\delta = g(\theta) - g_z(\theta)$ where $g(\theta)$ is the exact gradient, significantly hinders the optimization process when large step sizes are used. This motivates us to analyze the effects of this gradient noise $\delta$ and understand how it impacts optimization performance.

To quantify how the gradient noise $\delta$ hurts the optimization process, we evaluate its effect on the generalization performance of the estimated gradients. Specifically, we measure whether the zeroth-order gradient estimate computed on one batch can effectively reduce the loss on other held-out batches. For a batch $\mathcal{B}_t = \{\mathcal{B}_t^1, \mathcal{B}_t^2\}$ with 32 data points, we use 16 samples to estimate the zeroth-order gradient $g_z(\theta; \mathcal{B}_t^1)$ on batch $\mathcal{B}_t^1$, and evaluate it on the remaining 16 held-out samples $\mathcal{B}_t^2$. The results are shown in Figure 2. Interestingly, we find a stark contrast in performance - while the estimated gradient $g_z(\theta; \mathcal{B}_t^1)$ can reliably reduce the loss on the same batch $\mathcal{B}_t^1$ it was computed on (90% success rate), it only manages to decrease the loss on the new held-out batch $\mathcal{B}_t^2$ around 50% of the time. This suggests that the zeroth-order gradient estimates suffer from overfitting or noise that makes them less generalizable to unseen data samples. The gradient noise $\delta$, while allowing descent on the current batch, appears to introduce errors that prevent reliable descent directions for unseen batches. Therefore, the noise $\delta$ can be seen as hurting the optimization process by degrading the generalization performance of the parameter updates.

Next, we aim to understand if this effect is uniform across all model parameters or if certain parameter groups are more vulnerable to noise corruption. We notice the nature of vanilla MeZO, where $\frac{\mathcal{L}(\theta+\epsilon z) - \mathcal{L}(\theta-\epsilon z)}{2\epsilon}$ is used to estimate the gradient, and all parameters share the same value of $\frac{\mathcal{L}(\theta+\epsilon z) - \mathcal{L}(\theta-\epsilon z)}{2\epsilon}$. This means not all parameters are optimized in the true gradient direction, which could be a limitation. To analyze this, we divide the parameters into different groups based on their magnitude - the top 20% largest weights are considered "large", while the bottom 20% are "small". Interestingly, our experiments reveal that the gradient noise $\delta$ hurts optimization more when added to large weights compared to small weights. As shown in Figure 2(c), when continuing training from the point where test accuracy drops (due to noise), we find that optimizing only the small weights can recover and further improve test accuracy. This suggests that small weights are less impacted by noise corruption and can generalize better. Therefore, the noise $\delta$ does not impact all parameters equally - it disproportionately disrupts the optimization of larger weights. Selectively optimizing smaller, noise-resilient weights may be a promising direction to mitigate the effects of gradient noise in zeroth-order optimization. In the next section, we will introduce the proposed Spare-MeZO algorithm, which can only select small weights to perturb and update weights.

---

**Algorithm 1** Sparse-MeZO (S-MeZO)

---

**Require:** $\boldsymbol{\theta}$ represents pre-trained LLM weight, $N$ is the number of layers in model, learning rate $\eta_t$, $s$ represents sparsification interval.
  Initialize random seed $s$
  Determine threshold $\boldsymbol{h} = h_1, \ldots, h_N$, of each layer with the sparsification interval
  **for** $t \leftarrow 1$ **to** $T$ **do**
    Sample Minibatch $\mathcal{B}$ from $X$ and random seed $s$.
    $\boldsymbol{m} \leftarrow \text{GetMask}(\boldsymbol{\theta_t}, \boldsymbol{h})$
    $\boldsymbol{\theta_t} \leftarrow \text{PerturbParameters}(\boldsymbol{\theta_t}, \epsilon, s, \boldsymbol{m})$
    $l^+ = \mathcal{L}(\boldsymbol{\theta_t}; \mathcal{B})$
    $\boldsymbol{\theta_t} \leftarrow \text{PerturbParameters}(\boldsymbol{\theta_t}, -2\epsilon, s, \boldsymbol{m})$
    $l^- = \mathcal{L}(\boldsymbol{\theta_t}; \mathcal{B})$
    $\boldsymbol{\theta_t} \leftarrow \text{PerturbParameters}(\boldsymbol{\theta}, \epsilon, s, \boldsymbol{m})$
    $\text{proj\_grad} \leftarrow (l^+ - l^-)/(2\epsilon)$
    Reset random seed $s$
    **for** $\theta_i \in \boldsymbol{\theta}$ **do**
      $z_i \sim \mathcal{N}(0, 1)$
      $\theta_i \leftarrow \theta_i - \eta_t * \text{proj\_grad} * m_i * z$
    **end for**
  **end for**

---

## 3.2 SPARSE-MEZO

Consider a labelled dataset $\mathcal{D} = \{(\boldsymbol{x}_i, \boldsymbol{y}_i)\}_{i \in [|\mathcal{D}|]}$ and let $\mathcal{L}(\boldsymbol{\theta}; \mathcal{B})$ denotes the loss on a mini-batch $\mathcal{B}$. We can define a sparse mask $\boldsymbol{m} \in \{0, 1\}^d$ to selectively sample the random noise $\boldsymbol{z} \in \mathbb{R}^d$ with $\boldsymbol{z} \sim \mathcal{N}(\boldsymbol{0}, \boldsymbol{I}_d)$ on the sub-net of pre-trained model. A sparsified version of random perturbation can be defined as $\hat{\boldsymbol{z}} \in \mathbb{R}^d$:

$$\hat{\boldsymbol{z}} = \boldsymbol{m} \odot \boldsymbol{z}. \tag{1}$$

Based on this sparse perturbation $\hat{\boldsymbol{z}}$, we can redefine MeZO algorithm on Section 2.2.1 as Sparse-MeZO. The main difference is from the estimated gradient $\boldsymbol{g}_{\hat{\boldsymbol{z}}}(\boldsymbol{\theta})$, which can be defined as :

$$\begin{aligned}
\boldsymbol{g}_{\hat{\boldsymbol{z}}}(\boldsymbol{\theta}) &= \frac{\mathcal{L}(\boldsymbol{\theta} + \epsilon\hat{\boldsymbol{z}}; \mathcal{B}) - \mathcal{L}(\boldsymbol{\theta} - \epsilon\hat{\boldsymbol{z}}; \mathcal{B})}{2\epsilon}\hat{\boldsymbol{z}} \\
&= \frac{\mathcal{L}(\boldsymbol{\theta} + \epsilon\boldsymbol{m} \odot \boldsymbol{z}; \mathcal{B}) - \mathcal{L}(\boldsymbol{\theta} - \epsilon\boldsymbol{m} \odot \boldsymbol{z}; \mathcal{B})}{2\epsilon}\hat{\boldsymbol{z}},
\end{aligned} \tag{2}$$

where $\epsilon$ represents the perturbation scale.

Based on the observations from our motivation, we can create a sparse mask, $\boldsymbol{m}$, determined by parameter magnitudes. Specifically, we only update parameters of smaller magnitude. These targeted parameters are defined as $\hat{\boldsymbol{\theta}} = \boldsymbol{m} \odot \boldsymbol{\theta}$. It's important to note that we still preserve the complete set of parameters, but we apply sparse perturbations and gradient estimations only to the selected ones. This approach allows us to integrate the sparse mask into the standard MeZO method as a straightforward, adaptable tool. Then, we will introduce when and how to calculate the mask.

- **Constant Mask: Setting the Mask Before Training.** We compare the parameter values to a threshold for each layer to set the mask before training begins. However, a significant downside of this approach is the extra memory required to store a sparse mask, which is as large as the pre-trained model itself. Our goal is for our method to enhance performance without using more GPU memory or causing extra overhead.
- **Dynamic Mask: Determining Mask at Each Iteration.** We can establish a threshold for each layer before training and then generate the mask by comparing parameter values to this threshold during each iteration. This method avoids the necessity of storing a large mask, $\boldsymbol{m}$.

In this paper, we'll employ a dynamic mask to choose which parameters to perturb and update, addressing the issue of memory constraints. In addition, we determine thresholds using a principled

sparsity-based approach. Specifically, we use a percentile-based method where the threshold is set based on a target sparsity level.

The pseudo-code is provided in Algorithm 1. This algorithm outlines that we first establish the threshold $h_i$ for each layer before beginning training. We then use GetMask (Algorithm 3) to compare each parameter against its threshold $h_i$ and create the mask $\boldsymbol{m}$. Following this, we introduce the function PerturbParameters (Algorithm 2) to generate a Gaussian noise sample $\boldsymbol{z} \sim \mathcal{N}(\boldsymbol{0}, \boldsymbol{I_d})$ and apply the mask $\boldsymbol{m}$ to produce a sparse perturbation $\hat{\boldsymbol{z}} = \boldsymbol{m} \odot \boldsymbol{z}$. With $\hat{\boldsymbol{z}}$, we perturb the current parameters $\boldsymbol{\theta_t}$ to get new parameters $\boldsymbol{\theta_t} + \epsilon \hat{\boldsymbol{z}}$ and $\boldsymbol{\theta_t} - \epsilon \hat{\boldsymbol{z}}$. This allows us to compute two distinct loss values: $l^+ = \mathcal{L}(\boldsymbol{\theta_t} + \epsilon \hat{\boldsymbol{z}})$ and $l^- = \mathcal{L}(\boldsymbol{\theta_t} - \epsilon \hat{\boldsymbol{z}})$. From these losses, we calculate the estimated sparse gradient $\boldsymbol{g_m}(\boldsymbol{\theta_t}) = \text{proj\_grad} * \hat{\boldsymbol{z}}$, where $\text{proj\_grad} = \frac{l^+ - l^-}{2\epsilon}$. Finally, this gradient can be used with a learning rate $\eta_t$ to update $\boldsymbol{\theta_t}$.

## 3.3 MEMORY-EFFICIENT IMPLEMENTATION OF SPARSE-MEZO

In this paper, our primary aim is to introduce an efficient method for fine-tuning language models using zeroth-order optimization, enhancing performance on downstream tasks. As outlined in Algorithm 1, our approach involves perturbing the parameters $\boldsymbol{\theta_t}$ twice to generate two distinct sets of parameters, $\boldsymbol{\theta'_t} = \boldsymbol{\theta_t} + \epsilon \boldsymbol{z}$ and $\boldsymbol{\theta''_t} = \boldsymbol{\theta_t} - \epsilon \boldsymbol{z}$. We then use the estimated gradient to update the original parameters $\boldsymbol{\theta_t}$. This step typically requires storing two separate sets of parameters, leading to increased memory usage during fine-tuning.

Recently proposed MeZO, conserves memory by saving random seeds $s$ and using it to resample $z$ for calculating $\boldsymbol{\theta'_t}, \boldsymbol{\theta''_t}$, and reconstructing $\boldsymbol{\theta_t}$ without needing extra memory. However, applying a sparse mask $\boldsymbol{m}$ for calculating sparse perturbation $\hat{\boldsymbol{z}}$ in MeZO poses a memory issue. We cannot simply reconstruct $\hat{\boldsymbol{z}}$ by saving the random seed because the sparse mask, determined by parameter magnitudes, changes when parameters are altered by the perturbation. To address this, we propose potential solutions for the memory issue.

**1-bit Quantization:** We can apply 1-bit quantization to store the mask $\boldsymbol{m}$, as it consists solely of 0s and 1s. However, this method still increases memory usage, which isn't our goal. As a solution, we introduce a novel, memory-saving approach for zeroth-order optimization that calculates the mask $\boldsymbol{m}$ on the fly during the forward pass.

**Calculating the Mask During the Forward Pass:** By computing the mask and perturb parameters in the forward pass, we eliminate the need to store perturbed parameters $\boldsymbol{\theta'_t}$ and $\boldsymbol{\theta''_t}$. This means we only have to keep the original parameters $\boldsymbol{\theta_t}$ throughout training. For vanilla implementation, we first need to calculate the perturbed parameters with mask $\boldsymbol{m}$: $\boldsymbol{\theta'_t} = \boldsymbol{\theta_t} + \epsilon \boldsymbol{m} \odot \boldsymbol{z}$. After that, we can use perturbed parameters $\boldsymbol{\theta'_t}$ to calculate the loss value $l^+$ with the forward process. For example, the output of layer $i$ can be defined as $\boldsymbol{y^{(i)}} = \boldsymbol{\theta'^{(i)}_t} \boldsymbol{x^{(i)}} + \boldsymbol{b^{(i)}}$. Noted that we need to save the vanilla parameters $\boldsymbol{\theta_t}$ and mask $\boldsymbol{m}$ for vanilla implementation. However, for our proposed efficient implementation, we only need to save vanilla parameters $\boldsymbol{\theta_t}$. More specially, we can calculate the mask $\boldsymbol{m^{(i)}}$ of layer $i$ during the forward process and then obtain the output of this layer: $\boldsymbol{y^{(i)}} = (\boldsymbol{\theta^{(i)}_t} + \epsilon m(\boldsymbol{\theta_t}) \boldsymbol{z^{(i)}}) \boldsymbol{x^{(i)}} + \boldsymbol{b^{(i)}}$, where $m(\cdot)$ represents the function GetMask to calculate mask $\boldsymbol{m^{(i)}}$. Then, we can release the memory of mask $\boldsymbol{m^{(i)}}$ and calculate the output and mask of the next layer.

## 4 EXPERIMENTS

Following a similar setting to MeZO, we evaluate the performance of our proposed method on SuperGLUE Wang et al. (2019). The experimental results show that our proposed method can achieve better performance while also attaining faster convergence.

## 4.1 EXPERIMENTAL SETTING

**Datasets.** To verify the performance gain of our proposed method, we conduct experiments on various fine-tuning tasks include SST-2 (Socher et al., 2013), RTE (Bentivogli et al., 2009; Dagan et al., 2005; Giampiccolo et al., 2007; Haim et al., 2006), BoolQ (Clark et al., 2019), WIC (Pilehvar &

Camacho-Collados, 2018), MultiRC (Khashabi et al., 2018)) and multi-class task COPA (Roemmele et al., 2011).

**Models.** We primarily use pre-trained LLaMA-7b Touvron et al. (2023) to evaluate the performance of our proposed method on downstream tasks. To further demonstrate our method's versatility, we also test it with Mistral-7B-v0.1 Jiang et al. (2023) and OPT-13b Zhang et al. (2022). We provide more details about the results in the Appendix E. Additionally, to examine our method's scalability, we evaluate it on larger models, such as LLaMA-30b.

**Baselines.** First, we compare our method to the vanilla MeZO to demonstrate how sparsification enhances MeZO's convergence speed and overall performance. Additionally, to show that our proposed S-MeZO effectively identifies and modifies crucial parameters, we contrast it with R-MeZO (a version of MeZO applying a random mask to select parameters for optimization). In addition, we also explore the impact of zero-shot optimization on improving a pre-trained language model's capabilities through experiments with zeroth-shot learning and in-context learning techniques (Brown et al., 2020). Lastly, to understand the performance gap between zeroth-order and first-order optimization in fine-tuning large language models, we present results from conventional full-parameter fine-tuning (FT) using the Adam optimizer, the most widely used method for such tasks. In addition, we also compare MeZO and its variants against LoRA, the most widely adopted PEFT method.

**Training Procedure.** We adopt most of the training hyperparameters from the standard MeZO, including dataset configuration, batch size, training epochs, epsilon value, and task prompts, with the key difference being a higher learning rate for S-MeZO due to updating only a subset of the parameters. The primary goal of our training is the next token prediction. For the dataset, we use MeZO's approach, randomly selecting 1,000 examples for training and testing the model on another set of 1,000 examples (Zhou et al., 2023). We perform the experiments using three different seeds and report the average of the outcomes. In addition, the total training steps for LLaMA, Mistral and OPT is 20,000 and we evaluate its performance on the test dataset every 100 steps.

| Model | Method | BoolQ | RTE | WIC | MultiRC | SST-2 | COPA | Average |
|---|---|---|---|---|---|---|---|---|
| **LLaMA-7b** | **Zero-Shot** | 65.1 | 49.5 | 50.6 | 55.8 | 79.7 | 59.7 | 60.1 |
| **LLaMA-7b** | **ICL** | 67.4 | 54.5 | 52.7 | 58.7 | 81.2 | 84.4 | 66.5 |
| **LLaMA-7b** | **LoRA** | 84.5 | 82.3 | 67.6 | 78.3 | 95.0 | 86.0 | 82.3 |
| **LLaMA-7b** | **FT** | 84.5 | 83.6 | 68.4 | 80.2 | 95.7 | 85.0 | 82.9 |
| **LLaMA-7b** | **MeZO** | 75.9 | 71.7 | 61.4 | 69.8 | 94.6 | 86.3 | 76.6 |
| **LLaMA-7b** | **MeZO - LoRA** | 77.9 | 74.9 | 60.8 | 72.6 | 95.0 | 84.3 | 77.6 |
| **LLaMA-7b** | **R-MeZO** | 76.9 | 75.2 | 62.1 | 68.1 | 94.6 | 84.3 | 76.9 |
| **LLaMA-7b** | **S-MeZO** | **80.9** | **80.7** | **64.9** | **73.3** | **95.0** | **86.7** | **80.3** |

Table 1: Accuracy of Fine-Tuning LLaMA-7b on SuperGLUE (1,000 examples). ICL: In-Context Learning, FT: full-parameter fine-tuning with Adam, R-MeZO: MeZO with Random Mask.

| Model | Method | BoolQ | RTE | WIC | MultiRC | SST-2 | COPA | Average |
|---|---|---|---|---|---|---|---|---|
| **Mistral-7b** | **Zero-Shot** | 69.3 | 55.2 | 50.0 | 57.1 | 55.5 | 84.0 | 61.85 |
| **Mistral-7b** | **ICL** | 76.7 | 78.0 | 61.4 | 71.3 | 94.6 | 90.0 | 78.66 |
| **Mistral-7b** | **LoRA** | 84.8 | 87.4 | 68.2 | 83.9 | 95.6 | 91.0 | 85.15 |
| **Mistral-7b** | **FT** | 86.7 | 87.1 | 71.2 | 86.1 | 95.6 | 91.2 | 86.31 |
| **Mistral-7b** | **MeZO** | 81.6 | 80.9 | 63.2 | 82.7 | 93.8 | 86.7 | 81.48 |
| **Mistral-7b** | **MeZO - LoRA** | 83.5 | 80.1 | 60.7 | 82.6 | 93.8 | 86.9 | 81.26 |
| **Mistral-7b** | **R-MeZO** | 84.0 | 78.7 | 63.2 | 83.1 | 92.4 | 84.1 | 80.91 |
| **Mistral-7b** | **S-MeZO** | **85.3** | **84.5** | **64.3** | **84.9** | **94.2** | **86.1** | **83.21** |

Table 2: Accuracy of Fine-Tuning Mistral-7b on SuperGLUE (1,000 examples). ICL: In-Context Learning, FT: full-parameter fine-tuning with Adam, R-MeZO: MeZO with Random Mask.

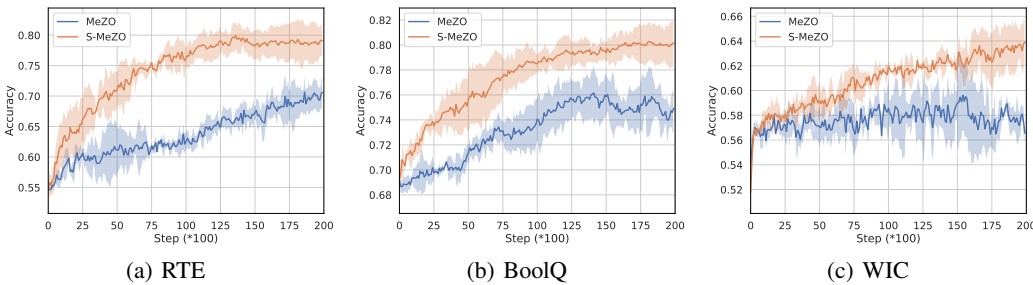

(a) RTE          (b) BoolQ          (c) WIC

Figure 3: Convergence Curves of Fine-Tuning LLaMA-7b with MeZO and Sparse-MeZO (S-MeZO) on (a) RTE, (b) BoolQ, (c) WIC tasks.

## 4.2 PERFORMANCE

To evaluate the performance of our proposed method S-MeZO, we initially tested it on the SuperGLUE benchmark using the LLaMA-7b model. The fine-tuning results, presented in Table 10, reveal that our S-MeZO method outperforms other zero-order (ZO) techniques like MeZO and R-MeZO. For instance, S-MeZO boosts MeZO's accuracy from $71.7\%$ to $80.7\%$ on RTE ($\uparrow 9\%$) and from $75.9\%$ to $80.9\%$ on BoolQ ($\uparrow 5\%$). Furthermore, all zeroth-order-based methods surpassed the performance of Zero-shot learning and in-context learning, demonstrating that zeroth-order optimization significantly enhances the pre-trained model's effectiveness on downstream tasks. Finally, we can find that S-MeZO significantly bridges the performance gap between zero-order and first-order optimization methods.

To further verify the generality of our proposed S-MeZO, we also evaluate it on Mistral-7B-v0.1. The performance is shown in Table 2. We can find that S-MeZO can consistently improve the performance of vanilla MeZO and narrow down the performance gap between zeroth-order optimization and first-order optimization. For example, S-MeZO can improve the accuracy of vanilla MeZO from $81.6$ to $85.3$ on BoolQ and then achieve a comparable performance with full fine-tuning.

## 4.3 CONVERGENCE RATE

To verify that S-MeZO converges faster than MeZO, we carried out multiple experiments for comparison. The accuracy over steps is plotted in Figure 3, which shows that S-MeZO can use fewer steps to achieve a better performance than vanilla MeZO. For example, S-MeZO only needs about 5,000 steps to achieve $70\%$ accuracy but vanilla MeZO needs 17,500 steps. Finally, S-MeZO can achieve about 3.5x speedup on RTE and 3x speedup on BoolQ.

| Method | SST-2 | RTE | BoolQ | WIC | MultiRC | COPA | Average |
|---|---|---|---|---|---|---|---|
| **FT** | 114.7 | 123.7 | 128.7 | 115.3 | 158.6 | 119.1 | 128.2 |
| **LoRA** | 15.7 | 19.5 | 25.5 | 16.1 | 34.2 | 23.1 | 22.4 |
| **MeZO** | 14.6 | 14.6 | 14.6 | 14.6 | 14.6 | 14.6 | 14.6 |
| **S-MeZO** | 28.3 | 28.3 | 28.3 | 28.3 | 28.3 | 28.3 | 28.3 |
| **S-MeZO-EI** | 14.6 | 14.6 | 14.6 | 14.6 | 14.6 | 14.6 | 14.6 |

Table 3: Memory Usage (batch size = 1) of Fine-Tuning LLaMA-7b on SuperGLUE (1,000 examples). EI represents the Efficient Implementation in section 3.3.

## 4.4 MEMORY USAGE

Table 3 shows the memory consumption for MeZO, S-MeZO, and traditional full-parameter fine-tuning of LLaMA-7b. The data reveal that S-MeZO does not require more memory than MeZO and

offers a substantial saving of roughly 12 times less GPU memory compared to full-parameter fine-tuning. For instance, S-MeZO with Efficient Implementation (S-MeZO-EI) cuts down the memory needed from $158.6$ GB for full tuning to just $14.6$ GB on MultiRC task. In addition, S-MeZO with efficient implementation can reduce the memory cost from $28.3$ GB of vanilla S-MeZO to $14.6$ GB across all five tasks, which also illustrates the efficiency of our proposed implementation method: Calculating the Mask During the Forward Pass. As a result, we can use only inference memory cost to fine-tune large language models.

## 4.5 Sparse Rate

For S-MeZO, we need to define the sparsity of the pre-trained model before starting to fine-tune it. To analyze the effects of sparsity value on the performance, we conduct experiments with various sparsity values (from $0.0$ to $0.85$). Figure 4 summarizes these experimental results with different sparsity values. We can find that a significant performance gain can be obtained when we use the sparsity value from $0.5$ to $0.8$. In addition, for most tasks, a sparsity value of $0.8$ or $0.75$ usually means a better performance. For example, S-MeZO can improve the accuracy from $71.7\%$ (when $r = 0.0$) to $82.3\%$ (when $r = 0.8$). It can also obtain a performance gain of $6.6\%$ for WIC (from $75.9\%$ to $82.5\%$).

| Model | Method | BoolQ | RTE | WIC |
|---|---|---|---|---|
| **LLaMA-7b** | **MeZO** | 75.9 | 71.7 | 61.4 |
| **LLaMA-7b** | **S-MeZO** | 80.9 | 80.7 | 64.9 |
| **LLaMA-30b** | **MeZO** | 83.8 | 76.9 | 63.3 |
| **LLaMA-30b** | **S-MeZO** | **85.7** | **82.1** | **67.3** |

Table 4: Accuracy of Fine-Tuning LLaMA-7b and LLaMA-30b on SuperGLUE (1,000 examples).

## 4.6 Scalability

In Table 10, we mainly introduce the performance of our methods on LLaMA-7b. A direct question is whether our proposed method can scale to larger language models. Therefore, in this section, we further explore our proposed method S-MeZO on LLaMA-30b. As shown in Table 4, we can see that the a larger model usually can obtain a better fine-tuned performance. For example, the accuracy on RTE with MeZO can be improved from $71.1\%$ on LLaMA-7b to $76.9\%$ on LLaMA-30b. Our method S-MeZO can further improve the performance on RTE to $82.1\%$ on LLaMA-30b. In addition, S-MeZO can further improve the accuracy on BoolQ to $85.7\%$ on LLaMA-30b.

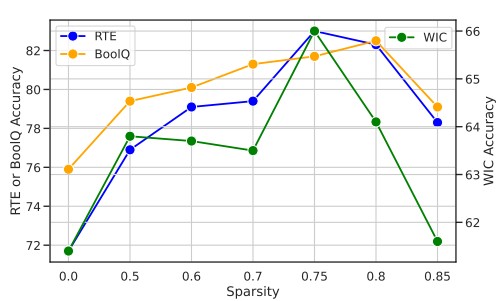

Figure 4: The effects of Sparsity for Fine-tuning LLaMA-7b with S-MeZO.

## 4.7 The Analysis about Efficient Implementation

In section 3.3, we present the efficient implementation of S-MeZO, which enables our proposed method to require only the inference memory cost for fine-tuning large language models. To analyze the actual GPU memory usage during the training process, we provide these results in Table 3. We can find that S-MeZO needs the same GPU memory for all five tasks, which can also save about $50\%$ memory compared to sparse-mezo. That also illustrates the efficiency of our proposed efficient implementation.

## 5 CONCLUSION

In this paper, we propose a novel memory-efficient zeroth-order fine-tuning method Sparse-MeZO, which can use a similar memory cost to the inference process. We evaluate the performance of fine-tuning LLaMA and OPT with Sparse-MeZO on SuperGULE benchmark and the experimental results illustrate that Sparse-MeZO can achieve a higher accuracy and faster convergence. Finally, we can fine-tune LLaMA-30b on a single A100 GPU.

**Limitation**: There is still a performance gap between our proposed method Sparse-MeZO and first-order fine-tuning methods. We plan to address these limitations and enhance Sparse-MeZO's capabilities in our future research.

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

# A APPENDIX

# B THE PROMPTS IN LLaMA AND OPT

| Dataset | Type | Prompt |
|---------|------|--------|
| SST-2 | cls. | {premise}
Does this mean that "{hypothesis}" is true? Yes or No?
Yes/No |
| RTE | cls. | Suppose "{premise}" Can we infer that "{hypothesis}"? Yes, No, or Maybe?
Yes/No/Maybe |
| BoolQ | cls. | {passage} {question} ?
Yes/No |
| WIC | cls. | Does the word "{word}" have the same meaning in these two sentences? Yes, No?
{sent1}
{sent2}
Yes/No |
| MultiRC | cls. | {paragraph}
Question: {question}
I found this answer "{answer}". Is that correct? Yes or No?
Yes/No |
| COPA | mch. | {premise} so/because {candidate} |

Table 5: The prompts of the datasets we used in our LLaMA experiments.

# C HYPERPARAMETERS

## C.1 HYPERPARAMETERS

We will introduce the hyperparameters searching grids in Table 7, which can help people reproduce our results.

| Experiment | Hyperparameters | Values |
|------------|-----------------|--------|
| MeZO | Batch size
Learning rate
$\epsilon$ | 16
$\{5e{-}7, 1e{-}6, 2e{-}6\}$
$1e{-}3$ |
| MeZO-Random | Batch size
Learning rate
$\epsilon$ | 16
$\{1e{-}6, 2e{-}6, 3e{-}6, 4e{-}6, 5e{-}6\}$
$1e{-}3$ |
| S-MeZO | Batch size
Learning rate
$\epsilon$ | 16
$\{1e{-}6, 2e{-}6, 3e{-}6, 4e{-}6, 5e{-}6\}$
$1e{-}3$ |
| FT with Adam | Batch size
Learning Rates | 8
$\{1e{-}5, 5e{-}5, 8e{-}5\}$ |

Table 6: The hyperparameter searching grids for LLaMA-7b experiments.

## C.2 THE SETTING OF THRESHOLD

We determine thresholds using a principled sparsity-based approach. Specifically, we use a percentile-based method where the threshold is set based on a target sparsity level. For example, with 80% sparsity, we sort the weight values of each layer and set the threshold at the 80th percentile. Importantly, this threshold is determined once before training begins and remains fixed throughout the optimization process.

| Experiment | Hyperparameters | Values |
|---|---|---|
| MeZO | Batch size | 16 |
| | Learning rate | $\{1e{-}8, 2e{-}8, 3e{-}8, 5e{-}8, 1e{-}7, 5e{-}7, 1e{-}6, 2e{-}6\}$ |
| | $\epsilon$ | $1e{-}3$ |
| MeZO-Random | Batch size | 16 |
| | Learning rate | $\{1e{-}6, 2e{-}6, 3e{-}6, 4e{-}6, 5e{-}6\}$ |
| | $\epsilon$ | $1e{-}3$ |
| S-MeZO | Batch size | 16 |
| | Learning rate | $\{1e{-}6, 2e{-}6, 3e{-}6, 4e{-}6, 5e{-}6\}$ |
| | $\epsilon$ | $1e{-}3$ |
| FT with Adam | Batch size | 8 |
| | Learning Rates | $\{1e{-}5, 5e{-}5, 8e{-}5\}$ |

Table 7: The hyperparameter searching grids for Mistral-7b experiments.

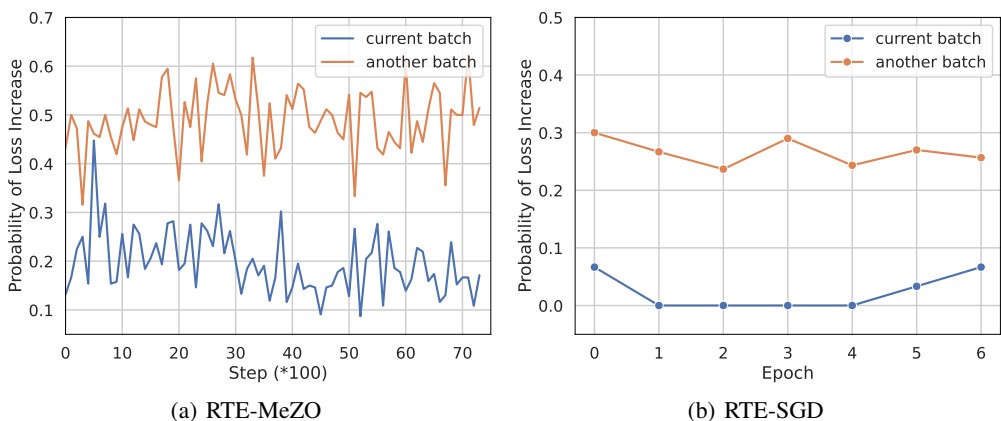

(a) RTE-MeZO

(b) RTE-SGD

Figure 5: (a) Probability of Loss Increase with MeZO on Different Batch. (b) Probability of Loss Increase with SGD on Different Batch. We calculate the probability of loss increment for each epoch.

We then introduce the sparsity of each task in SuperGULU when we fine-tune LLaMA-7b. The setting is shown in the Table 8.

| Method | SST-2 | RTE | BoolQ | WIC | MultiRC |
|---|---|---|---|---|---|
| **LLaMA + Sparse MeZO** | 0.70 | 0.75 | 0.80 | 0.80 | 0.80 |
| **Mistral + Sparse MeZO** | 0.70 | 0.60 | 0.60 | 0.70 | 0.60 |

Table 8: Sparsity in SuperGULU when we fine-tune LLaMA-7b and Mistral.

## D   COMPARISON BETWEEN MEZO AND SGD

## E   THE EXPERIMENTAL RESULTS ON OPT

We also provide the experimental results on OPT. As shown in the Table 9, Sparse MeZO can consistently improve the performance of vanilla MeZO on the three tasks of SuperGULE.

| Model | Method | BoolQ | RTE | WIC |
|--------|----------|-------|------|------|
| **OPT-13b** | **Zero Shot** | 59.0 | 59.6 | 55.0 |
| **OPT-13b** | **ICL** | 66.9 | 62.1 | 50.5 |
| **OPT-13b** | **MeZO** | 72.1 | 75.5 | 62.2 |
| **OPT-13b** | **R-MeZO** | 72.3 | 75.2 | 61.7 |
| **OPT-13b** | **S-MeZO** | **73.8** | **77.6** | **63.7** |

Table 9: Accuracy of Fine-Tuning OPT on SuperGLUE (1,000 examples). ICL: In-Context Learning, R-MeZO: MeZO with Random Mask.

## F    CONVERGENCE ANALYSIS OF SPARSE-MEZO

In this section, we will explain why Sparse-MeZO can accelerate the convergence, which is based on the theory from (Ohta et al., 2020). We can define a sub-network in pre-trained large language models, which is determined by the sparse mask $m$. The main idea of our proof is that if we follow the updated role in Sparse-MeZO, the gradient norm on the sub-network can be smaller than $\sigma^2$ after $\mathcal{O}(\frac{\hat{d}L}{\sigma^2})$ steps, where $\hat{d}$ is the number of parameters in the sub-network. Therefore, ZO can use fewer steps to converge when we only focus on a sub-network. Some related work has illustrated that only tuning the sub-network can achieve comparable performance, which will be empirically verified in our experiments.

Firstly, we assume the loss function $\mathcal{L}(\boldsymbol{\theta}; \boldsymbol{x})$ is Lipschitz Continuous:

**Assumption 1** (Lipschitz Continuous)**.**

$$\|\nabla\mathcal{L}(\boldsymbol{\theta}; \boldsymbol{x}) - \nabla\mathcal{L}(\boldsymbol{\theta}', \boldsymbol{x})\| \leq \frac{L(l)}{2}\|\boldsymbol{\theta} - \boldsymbol{\theta}'\|^2, \tag{3}$$

where $\nabla\mathcal{L}(\boldsymbol{\theta}; \boldsymbol{x})$ denotes the true first-order gradient of $\boldsymbol{\theta}$ on $\boldsymbol{x}$ and $L(l)$ represents the Lipschitz constant of $\mathcal{L}(\cdot)$. Given $\mathcal{L}_{\hat{z}}(\boldsymbol{\theta}) = \mathbb{E}_{\hat{z}}[\mathcal{L}(\boldsymbol{\theta} + \epsilon\hat{z})]$ and the above Assumption 1, we can obtain the relationship between sparse gradient $\widehat{\nabla}_{\boldsymbol{\theta}}\mathcal{L}_{\hat{z}}(\boldsymbol{\theta})$ and the expectation of estimated sparse ZO gradient $g_{\hat{z}}(\boldsymbol{\theta})$:

**Lemma 1.** *ZO gradient $g_{\hat{z}}(\boldsymbol{\theta})$ is unbiased estimation of $\widehat{\nabla}_{\boldsymbol{\theta}}\mathcal{L}_{\hat{z}}(\boldsymbol{\theta})$:*

$$\begin{aligned}
\widehat{\nabla}_{\boldsymbol{\theta}}\mathcal{L}_{\hat{z}}(\boldsymbol{\theta}) &= \boldsymbol{m} \odot \nabla_{\boldsymbol{\theta}}\mathcal{L}_{\hat{z}}(\boldsymbol{\theta}) \\
&= \boldsymbol{m} \odot \nabla_{\boldsymbol{\theta}}\mathbb{E}_{\hat{z}}[\mathcal{L}(\boldsymbol{\theta} + \epsilon\hat{z})] \\
&= \mathbb{E}_{\hat{z}}[\frac{\mathcal{L}(\boldsymbol{\theta} + \epsilon\hat{z}) - \mathcal{L}(\boldsymbol{\theta} - \epsilon\hat{z})}{2\epsilon}\hat{z}] \\
&= \mathbb{E}_{\hat{z}}[g_{\hat{z}}(\boldsymbol{\theta})],
\end{aligned} \tag{4}$$

where $g_{\hat{z}}(\boldsymbol{\theta}) = \frac{\mathcal{L}(\boldsymbol{\theta} + \epsilon\hat{z}) - \mathcal{L}(\boldsymbol{\theta} - \epsilon\hat{z})}{2\epsilon}\hat{z}$. We can find that $g_{\hat{z}}(\boldsymbol{\theta})$ is unbiased estimation of $\widehat{\nabla}_{\boldsymbol{\theta}}\mathcal{L}_{\hat{z}}(\boldsymbol{\theta})$. Then, based on the equation $\widehat{\nabla}_{\boldsymbol{\theta}}\mathcal{L}_{\hat{z}}(\boldsymbol{\theta}) = \mathbb{E}_{\hat{z}}[g_z(\boldsymbol{\theta})]$ in Lemma 1, we can use the distance $\|\widehat{\nabla}_{\boldsymbol{\theta}}\mathcal{L}_{\hat{z}}(\boldsymbol{\theta}) - \nabla_{\boldsymbol{\theta}}\mathcal{L}_m(\boldsymbol{\theta})\|$ to analyze the the relationship between the true sparse gradient $\nabla_{\boldsymbol{\theta}}\mathcal{L}_m(\boldsymbol{\theta}) = \boldsymbol{m} \odot \nabla_{\boldsymbol{\theta}}\mathcal{L}(\boldsymbol{\theta})$ and sparse gradient $\widehat{\nabla}_{\boldsymbol{\theta}}\mathcal{L}_{\hat{z}}(\boldsymbol{\theta})$:

**Lemma 2.** *Let $\mathcal{L}$ be Lipschitz Continuous, we have:*

$$\|\nabla_{\boldsymbol{\theta}}\mathcal{L}_m(\boldsymbol{\theta})\|^2 \leq 2\|\widehat{\nabla}_{\boldsymbol{\theta}}\mathcal{L}_{\hat{z}}(\boldsymbol{\theta})\|^2 + \frac{\epsilon^2 L^2(l)}{2}(\hat{d} + 4)^3. \tag{5}$$

where $\nabla_{\boldsymbol{\theta}}\mathcal{L}_m(\boldsymbol{\theta}) = \boldsymbol{m} \odot \nabla_{\boldsymbol{\theta}}\mathcal{L}(\boldsymbol{\theta})$, $\hat{d} = \sum_{i=1}^{i=d} m_i$ is the number of selected parameters in mask $\boldsymbol{m}$, $L(l)$ represents the Lipschitz constant. Finally, we can obtain the convergence rate of Sparse-MeZO.

**Theorem 1.** *Assuming a sequence of generated parameters $\{\boldsymbol{\theta_t}\}_{t\geq 0}$ in Sparse-MeZO. We can have:*

$$\mathbb{E}_{\hat{z},x}[\|\nabla_{\boldsymbol{\theta}}\mathcal{L}_m(\boldsymbol{\theta_T})\|^2] \leq \sigma^2 \tag{6}$$

*for any $T = \mathcal{O}(\frac{\hat{d}L}{\sigma^2})$*

where $L(l) \leq L$ for all $\mathcal{L}(\boldsymbol{\theta_t})$. This theorem illustrates that the presence of pronounced sparsity patterns, along with the smoothness of the objective function, can significantly enhance the rate of convergence, potentially achieving a linear acceleration.

## G  THE PROOF OF LEMMA 1

Let $\mathcal{L}_z(\theta)$ be the expectation of $\mathcal{L}(\theta + \epsilon m \odot z)$:

$$\begin{aligned}
\mathcal{L}_{\hat{z}}(\theta) :&= \mathbb{E}_z[\mathcal{L}(\theta + \epsilon m \odot z)]\\
&= \mathbb{E}_{\hat{z}}[\mathcal{L}(\theta + \epsilon\hat{z})]
\end{aligned} \tag{7}$$

We can obtain the Lemma:

$$\begin{aligned}
\widehat{\nabla}_{\theta}\mathcal{L}_{\hat{z}}(\theta) &= m \odot \nabla_{\theta}\mathcal{L}_{\hat{z}}(\theta)\\
&= m \odot \mathbb{E}_z[\nabla_{\theta}\mathcal{L}(\theta + \epsilon m \odot z)]\\
&= \mathbb{E}_z\Big[\frac{\mathcal{L}(\theta + \epsilon m \odot z) - \mathcal{L}(\theta - \epsilon m \odot z)}{2\epsilon} m \odot z\Big]\\
&= \mathbb{E}_{\hat{z}}\Big[\frac{\mathcal{L}(\theta + \epsilon\hat{z}) - \mathcal{L}(\theta - \epsilon\hat{z})}{2\epsilon}\hat{z}\Big]
\end{aligned} \tag{8}$$

Proof:

$$\begin{aligned}
\widehat{\nabla}_{\theta}\mathcal{L}_{\hat{z}}(\theta) &= \widehat{\nabla}_{\theta}\mathbb{E}_{\hat{z}}[\mathcal{L}(\theta + \epsilon\hat{z})]\\
&= \widehat{\nabla}_{\theta}\int_{\hat{z}}\mathrm{pdf}_{\hat{z}}(z)\mathcal{L}(\theta + \epsilon z)dz\\
&= m \odot \nabla_{\theta}\int_{\hat{z}}\mathrm{pdf}_{\hat{z}}(z)\mathcal{L}(\theta + \epsilon z)dz\\
&= m \odot \int_{\hat{z}}\nabla_{\theta}\mathrm{pdf}_{\hat{z}}(z)\mathcal{L}(\theta + \epsilon z)dz\\
&= \frac{1}{k}m \odot \int_{\hat{z}}\nabla_{\theta}e^{-\frac{1}{2}\|z\|^2}\mathcal{L}(\theta + \epsilon z)dz\\
&= \frac{1}{k}m \odot \int_{\hat{y}}\nabla_{\theta}e^{-\frac{1}{2}\|\frac{y-\theta}{\epsilon}\|^2}\mathcal{L}(y)\frac{1}{\epsilon^n}dy\\
&= \frac{1}{k}m \odot \int_{\hat{y}}\frac{y-\theta}{\epsilon^2}e^{-\frac{1}{2\epsilon^2}\|y-\theta\|^2}\mathcal{L}(y)\frac{1}{\epsilon^n}dy\\
&= \frac{1}{k}m \odot \int_{\hat{z}}\frac{z}{\epsilon}e^{-\frac{1}{2}\|z\|^2}\mathcal{L}(\theta + \epsilon z)dz\\
&= m \odot \int_{\hat{z}}\mathrm{pdf}_{\hat{z}}(z)\mathcal{L}(\theta + \epsilon z)\frac{z}{\epsilon}dz\\
&= \mathbb{E}_{\hat{z}}\Big[m \odot \frac{\mathcal{L}(\theta + \epsilon\hat{z})}{\epsilon}\hat{z}\Big]\\
&= \mathbb{E}_{\hat{z}}\Big[\frac{\mathcal{L}(\theta + \epsilon\hat{z})}{\epsilon}\hat{z}\Big]
\end{aligned} \tag{9}$$

where we can define $y = \theta + \epsilon z$, $\hat{y} = \theta + \epsilon m \odot z$, $k = \sqrt{(2\pi)^{\hat{d}}}$ and $\hat{d}$ is the number of 1 in $m$.

Therefore, we can obtain the gradient $\nabla_{\theta}\mathcal{L}_m(\theta)$ is equal to $\mathbb{E}_{\hat{z}}[\frac{\mathcal{L}(\theta+\epsilon\hat{z})}{\epsilon}\hat{z}]$.

In addition, we will prove $\mathbb{E}_{\hat{z}}\left[\frac{\mathcal{L}(\theta+\epsilon\hat{z})}{\epsilon}\hat{z}\right]$ is also equal to $\mathbb{E}_{\hat{z}}\left[\frac{\mathcal{L}(\theta+\epsilon\hat{z})-L(\theta)}{\epsilon}\hat{z}\right]$:

$$
\begin{aligned}
&\mathbb{E}_{\hat{z}}\left[\frac{\mathcal{L}(\theta+\epsilon\hat{z})-\mathcal{L}(\theta)}{\epsilon}\hat{z}\right]\\
&=\frac{1}{k}\int_{\hat{z}}\frac{\mathcal{L}(\theta+\epsilon z)-\mathcal{L}(\theta)}{\epsilon}ze^{-\frac{1}{2}\|z\|^2}dz\\
&=\frac{1}{k}\int_{\hat{\epsilon}}\frac{\mathcal{L}(\theta+\epsilon z)}{\epsilon}ze^{-\frac{1}{2}\|z\|^2}dz-\frac{\mathcal{L}(\theta)}{\epsilon k}\int_{\hat{z}}ze^{-\frac{1}{2}\|z\|^2}dz\\
&=\mathbb{E}_{\hat{z}}\left[\frac{\mathcal{L}(\theta+\epsilon\hat{z})}{\epsilon}\hat{z}\right]
\end{aligned}
\tag{10}
$$

After that, we can get the relationship between $\mathbb{E}_{\hat{z}}\left[\frac{\mathcal{L}(\theta)-\mathcal{L}(\theta-\epsilon\hat{z})}{\epsilon}\hat{z}\right]$ and $\mathbb{E}_{\hat{z}}\left[\frac{\mathcal{L}(\theta+\epsilon\hat{z})}{\epsilon}\hat{z}\right]$:

$$
\begin{aligned}
\mathbb{E}_{\hat{z}}\left[\frac{\mathcal{L}(\theta)-\mathcal{L}(\theta-\epsilon\hat{z})}{\epsilon}\hat{z}\right]&=\mathbb{E}_{\hat{z}}\left[\frac{\mathcal{L}(\theta+\epsilon(-\hat{z}))-\mathcal{L}(\theta)}{\epsilon}(-\hat{z})\right]\\
&=\mathbb{E}_{\hat{z}}\left[\frac{\mathcal{L}(\theta+\epsilon\hat{z}-\mathcal{L}(\theta))}{\epsilon}\hat{z}\right]\\
&=\mathbb{E}_{\hat{z}}\left[\frac{\mathcal{L}(\theta+\epsilon\hat{z})}{\epsilon}\hat{z}\right].
\end{aligned}
\tag{11}
$$

Based on the Equation 10 and Equation 11, we can obtain:

$$
\begin{aligned}
&\mathbb{E}_{\hat{z}}\left[\frac{\mathcal{L}(\theta+\epsilon\hat{z})-\mathcal{L}(\theta-\epsilon\hat{z})}{2\epsilon}\hat{z}\right]\\
&=\frac{1}{2}\left(\mathbb{E}_{\hat{z}}\left[\frac{\mathcal{L}(\theta+\epsilon\hat{z})}{\epsilon}\hat{z}-\frac{\mathcal{L}(\theta)}{\epsilon}\hat{z}+\frac{\mathcal{L}(\theta)}{\epsilon}\hat{z}-\frac{\mathcal{L}(\theta-\epsilon\hat{z})}{\epsilon}\hat{z}\right]\right)\\
&=\frac{1}{2}\left(\mathbb{E}_{\hat{z}}\left[\frac{\mathcal{L}(\theta+\epsilon\hat{z})-\mathcal{L}(\theta)}{\epsilon}\hat{z}\right]+\mathbb{E}_{\hat{z}}\left[\frac{\mathcal{L}(\theta)-\mathcal{L}(\theta-\epsilon\hat{z})}{\epsilon}\hat{z}\right]\right)\\
&=\frac{1}{2}\left(\mathbb{E}_{\hat{z}}\left[\frac{\mathcal{L}(\theta+\epsilon\hat{z})}{\epsilon}\hat{z}\right]+\mathbb{E}_{\hat{z}}\left[\frac{\mathcal{L}(\theta+\epsilon\hat{z})}{\epsilon}\hat{z}\right]\right)\\
&=\mathbb{E}_{\hat{z}}\left[\frac{\mathcal{L}(\theta+\epsilon\hat{z})}{\epsilon}\hat{z}\right]\\
&=\widehat{\nabla}_{\theta}\mathcal{L}_{\hat{z}}(\theta)
\end{aligned}
\tag{12}
$$

Finally, we can obtain the relationship between $\mathbb{E}_{\hat{z}}\left[\frac{\mathcal{L}(\theta+\epsilon\hat{z})-\mathcal{L}(\theta-\epsilon\hat{z})}{2\epsilon}\hat{z}\right]$ and $\widehat{\nabla}_{\theta}\mathcal{L}_{\hat{z}}(\theta)$ and finish the proof.

## H   THE PROOF OF LEMMA 2

$$
\|\nabla_{\boldsymbol{\theta}}\mathcal{L}_m(\boldsymbol{\theta})\|^2\leq 2\|\widehat{\nabla}_{\boldsymbol{\theta}}\mathcal{L}_{\hat{z}}(\boldsymbol{\theta})\|^2+\frac{\epsilon^2 L^2(l)}{2}(\hat{d}+4)^3.
\tag{13}
$$

Proof:

We can first define the distance between $\widehat{\nabla}_{\boldsymbol{\theta}}\mathcal{L}_{\hat{z}}(\boldsymbol{\theta})=\mathbb{E}_{\hat{z}}[\boldsymbol{g}_{\hat{z}}(\boldsymbol{\theta})]$ and sparse FO gradient $\nabla\mathcal{L}_m(\theta)$ as:

$$
\begin{aligned}
\|\widehat{\nabla}_\theta \mathcal{L}_{\hat{z}}(\theta) &- \nabla_\theta \mathcal{L}_m(\theta)\| \\
&= \|\frac{1}{k}\int_z (\frac{\mathcal{L}(\theta+\epsilon z) - \mathcal{L}(\theta-\epsilon z)}{2\epsilon} - \langle \nabla_\theta \mathcal{L}_m(\theta), z\rangle)z e^{-\frac{1}{2}\|z\|^2} d\hat{z}\| \\
&= \|\frac{1}{k}\int_z (\frac{\mathcal{L}(\theta+\epsilon z) - \mathcal{L}(\theta)}{\epsilon} - \langle m\odot\nabla_\theta\mathcal{L}(\theta), z\rangle)z e^{-\frac{1}{2}\|z\|^2} d\hat{z}\| \\
&\leq \frac{1}{k\epsilon}\int_z |\mathcal{L}(\theta+\epsilon z) - \mathcal{L}(\theta) - \epsilon\langle\nabla_\theta\mathcal{L}(\theta),\epsilon\rangle|\|m\odot z\|e^{-\frac{1}{2}\|z\|^2}d\hat{z} \\
&\leq \frac{\epsilon L(l)}{2k}\int_\epsilon \|z\|^2\|m\odot z\|e^{-\frac{1}{2}\|z\|^2}d\hat{z} \\
&= \frac{\epsilon L(l)}{2}\mathbb{E}_{\hat{z}}[\|\hat{z}\|^3] \\
&\leq \frac{\epsilon L(l)}{2}(\hat{d}+3)^{\frac{3}{2}}
\end{aligned}
\tag{14}
$$

where $\hat{d}$ is the number of selected parameters with mask $m$. In addition, $\|a+b\|^2 \leq 2\|a\|^2 + 2\|b\|^2$, we can define $a = a - b$ and obtain that $\|a\|^2 \leq 2\|a-b\|^2 + 2\|b\|^2$. Let $a = \nabla_\theta\mathcal{L}_m(\theta)$ and $b = \widehat{\nabla}_\theta\mathcal{L}_{\hat{z}}(\theta)$, we can obtain:

$$
\begin{aligned}
\|\nabla_\theta\mathcal{L}_m(\theta)\|^2 &\leq 2\|\widehat{\nabla}_\theta\mathcal{L}_{\hat{z}}(\theta) - \nabla_\theta\mathcal{L}_m(\theta)\|^2 + 2\|\widehat{\nabla}_\theta\mathcal{L}_{\hat{z}}(\theta)\|^2 \\
&\leq \frac{\epsilon^2 L^2(l)}{2}(\hat{d}+3)^3 + 2\|\widehat{\nabla}_\theta\mathcal{L}_{\hat{z}}(\theta)\|^2 \\
&\leq \frac{\epsilon^2 L^2(l)}{2}(\hat{d}+4)^3 + 2\|\widehat{\nabla}_\theta\mathcal{L}_{\hat{z}}(\theta)\|^2
\end{aligned}
\tag{15}
$$

## I  THE PROOF OF THEOREM 1

Proof:

$$
\begin{aligned}
\mathcal{L}_{\hat{z}}(\theta) - \mathcal{L}(\theta) &= \mathbb{E}_{\hat{z}}[\mathcal{L}(\theta+\epsilon\hat{z}) - \mathcal{L}(\theta)] \\
&= \mathbb{E}_{\hat{z}}[\mathcal{L}(\theta+\epsilon\hat{z}) - \mathcal{L}(\theta) - \epsilon\langle\nabla\mathcal{L}(\theta),\hat{z}\rangle] \\
&= \frac{1}{k}\int_{\hat{z}}[\mathcal{L}(\theta+\epsilon z) - \mathcal{L}(\theta) - \epsilon\langle\nabla\mathcal{L}(\theta),z\rangle]e^{-\frac{1}{2}\|z\|^2}dz \\
&\leq \frac{1}{k}\int_{\hat{z}}\frac{\epsilon^2 L(l)}{2}\|z\|^2 e^{-\frac{1}{2}\|z\|^2}dz \\
&= \frac{\epsilon^2 L(l)}{2}\mathbb{E}_{\hat{z}}[\|\hat{z}\|^2] \\
&\leq \frac{\epsilon^2 L(l)}{2}\hat{d}
\end{aligned}
\tag{16}
$$

The first inequality holds because Lipschitz Continuous: $|\mathcal{L}(\theta') - \mathcal{L}(\theta) - \langle\nabla\mathcal{L}(\theta),\theta'-\theta\rangle| \leq \frac{L(l)}{2}\|\theta'-\theta\|^2$, where $\theta' = \theta + \epsilon z$. The second inequality holds because $\mathbb{E}_{\hat{z}}[\|\hat{z}\|^2] = \hat{d}$, where $\hat{d}$ is the number of 1 in mask $m$.

$$
\begin{aligned}
[(\mathcal{L}_{\hat{z}}(\theta) &- \mathcal{L}(\theta)) - (\mathcal{L}_{\hat{z}}(\theta+\epsilon\hat{z}) - \mathcal{L}(\theta+\epsilon\hat{z}))]^2 \\
&\leq 2[\mathcal{L}_{\hat{z}}(\theta) - \mathcal{L}(\theta)]^2 + 2[\mathcal{L}_{\hat{z}}(\theta+\epsilon\hat{z}) - \mathcal{L}(\theta+\epsilon\hat{z})]^2 \\
&\leq \frac{\epsilon^4 L^2(l)}{2}\hat{d}^2 + \frac{\epsilon^4 L^2(l)}{2}\hat{d}^2 \\
&= \epsilon^4 L^2(l)\hat{d}^2
\end{aligned}
\tag{17}
$$

The first inequality is due to $\|\boldsymbol{a} + \boldsymbol{b}\|^2 \le 2\|\boldsymbol{a}\|^2 + 2\|\boldsymbol{b}\|^2$, where $\boldsymbol{a} = \mathcal{L}_{\hat{z}}(\theta) - \mathcal{L}(\theta), \boldsymbol{b} = \mathcal{L}_{\hat{z}}(\theta + \epsilon\hat{z}) - \mathcal{L}(\theta + \epsilon\hat{z})$. The second inequality is due to the Equation 16.

$$
\begin{aligned}
[\mathcal{L}_{\hat{z}}(\theta + \epsilon\hat{z}) - \mathcal{L}_{\hat{z}}(\theta)]^2 &\le 2[\mathcal{L}_{\hat{z}}(\theta + \epsilon\hat{z}) - \mathcal{L}_{\hat{z}}(\theta) - \epsilon\langle\widehat{\nabla}_\theta\mathcal{L}_{\hat{z}}(\theta), \hat{z}\rangle]^2 + 2[\epsilon\langle\widehat{\nabla}_\theta\mathcal{L}_{\hat{z}}(\theta), \hat{z}\rangle]^2 \\
&\le \frac{\epsilon^4 L^2(l)}{2}\|\hat{z}\|^4 + 2\epsilon^2\langle\widehat{\nabla}_\theta\mathcal{L}_{\hat{z}}(\theta), \hat{z}\rangle^2 \\
&\le \frac{\epsilon^4 L^2(l)}{2}\|\hat{z}\|^4 + 2\epsilon^2\|\widehat{\nabla}_\theta\mathcal{L}_{\hat{z}}(\theta)\|^2\|\hat{z}\|^2
\end{aligned}
\tag{18}
$$

The first inequality is due to $\|\boldsymbol{a} + \boldsymbol{b}\|^2 \le 2\|\boldsymbol{a}\|^2 + 2\|\boldsymbol{b}\|^2$. The second inequality holds because Lipschitz Continuous: $|\mathcal{L}(\theta') - \mathcal{L}(\theta) - \langle\nabla\mathcal{L}(\theta), \theta' - \theta\rangle| \le \frac{L(l)}{2}\|\theta' - \theta\|^2$, where $\theta' = \theta + \epsilon\hat{z}$.

$$
\begin{aligned}
&[\mathcal{L}(\theta + \epsilon\hat{z}) - \mathcal{L}(\theta)]^2 \\
&\le 2[(\mathcal{L}_{\hat{z}}(\theta) - \mathcal{L}(\theta)) - (\mathcal{L}_{\hat{z}}(\theta + \epsilon\hat{z}) - \mathcal{L}(\theta + \epsilon\hat{z}))]^2 + 2[\mathcal{L}_{\hat{z}}(\theta + \epsilon\hat{z}) - \mathcal{L}_{\hat{z}}(\theta)]^2 \\
&\le 2\epsilon^4 L^2(l)\hat{d}^2 + \epsilon^4 L^2(l)\|\hat{z}\|^4 + 4\epsilon^2\|\widehat{\nabla}_\theta\mathcal{L}_{\hat{z}}(\theta)\|^2\|\hat{z}\|^2
\end{aligned}
\tag{19}
$$

The first inequality is due to $\|\boldsymbol{a} + \boldsymbol{b}\|^2 \le 2\|\boldsymbol{a}\|^2 + 2\|\boldsymbol{b}\|^2$. The last inequality holds because Equation 17 and Equation 18.

$$
\begin{aligned}
\mathbb{E}_{z,x}[\|g_{\hat{z}}(\theta)\|^2] &= \mathbb{E}_{\hat{z}}[\|\frac{\mathcal{L}(\theta + \epsilon\hat{z}) - \mathcal{L}(\theta - \epsilon\hat{z})}{2\epsilon}\hat{z}\|^2] \\
&= \mathbb{E}_{\hat{z}}[\|\frac{\mathcal{L}(\theta + \epsilon\hat{z}) - \mathcal{L}(\theta)}{2\epsilon}\hat{z} + \frac{\mathcal{L}(\theta) - \mathcal{L}(\theta - \epsilon\hat{z})}{2\epsilon}\hat{z}\|^2] \\
&\le \mathbb{E}_{\hat{z}}[2\|\frac{\mathcal{L}(\theta + \epsilon\hat{z}) - \mathcal{L}(\theta)}{2\epsilon}\hat{z}\|^2 + 2\|\frac{\mathcal{L}(\theta) - \mathcal{L}(\theta - \epsilon\hat{z})}{2\epsilon}\hat{z}\|^2] \\
&= \mathbb{E}_{\hat{z}}[\frac{1}{2\epsilon^2}[\mathcal{L}(\theta + \epsilon\hat{z}) - \mathcal{L}(\theta)]^2 \cdot \|\hat{z}\|^2 + \frac{1}{2\epsilon^2}[\mathcal{L}(\theta) - \mathcal{L}(\theta - \epsilon\hat{z})]^2 \cdot \|\hat{z}\|^2] \\
&\le \mathbb{E}_{\hat{z}}[2\epsilon^2 L^2(l)\hat{d}^2\|\hat{z}\|^2 + \epsilon^2 L^2(l)\|\hat{z}\|^6 + 4\|\widehat{\nabla}\mathcal{L}_{\hat{z}}(\theta)\|^2\|\hat{z}\|^4] \\
&\le 2\epsilon^2 L^2(l)\hat{d}^3 + \epsilon^2 L^2(l)(\hat{d} + 6)^3 + 4(\hat{d} + 4)^2\|\widehat{\nabla}\mathcal{L}_{\hat{z}}(\theta)\|^2 \\
&\le 3\epsilon^2 L^2(l)(\hat{d} + 4)^3 + 4(\hat{d} + 4)^2\|\widehat{\nabla}\mathcal{L}_{\hat{z}}(\theta)\|^2
\end{aligned}
\tag{20}
$$

The first inequality holds because $\|\boldsymbol{a} + \boldsymbol{b}\|^2 \le 2\|\boldsymbol{a}\|^2 + 2\|\boldsymbol{b}\|^2$, where $\boldsymbol{a} = \frac{\mathcal{L}(\theta + \epsilon\hat{z}) - \mathcal{L}(\theta)}{2\epsilon}\hat{z}$, $\boldsymbol{b} = \frac{\mathcal{L}(\theta) - \mathcal{L}(\theta - \epsilon\hat{z})}{2\epsilon}\hat{z}$. The second inequality is due to the Equation 19. The third inequality holds because $\mathbb{E}_{\hat{z}}[\|\hat{z}\|^2] = \hat{d}$, $\mathbb{E}_{\hat{z}}[\|\hat{z}\|^p] \le (\hat{d} + p)^{\frac{p}{2}}$ for $p \ge 2$. The last inequality holds because $2\hat{d}^3 + (\hat{d} + 6)^3 \le 3(\hat{d} + 4)^3$.

Based on the assumption about Lipschitz Continuous, we can obtain: $|\mathcal{L}(\theta_{t+1}) - \mathcal{L}(\theta_t) - \langle\nabla\mathcal{L}(\theta_t), \theta_{t+1} - \theta_t\rangle| \le \frac{L(l)}{2}\|\theta_{t+1} - \theta_t\|^2$.

Then, we can obtain:

$$
\mathcal{L}_{\hat{z}}(\theta_{t+1}) - \mathcal{L}_{\hat{z}}(\theta_t) - \langle\widehat{\nabla}\mathcal{L}_{\hat{z}}(\theta_t), \theta_{t+1} - \theta_t\rangle \le |\mathcal{L}_{\hat{z}}(\theta_{t+1}) - \mathcal{L}_{\hat{z}}(\theta_t) - \langle\widehat{\nabla}\mathcal{L}_{\hat{z}}(\theta_t), \theta_{t+1} - \theta_t\rangle| \le \frac{L(l)}{2}\|\theta_{t+1} - \theta_t\|^2
\tag{21}
$$

Based on the equation, we can follow the update rule: $\theta_{t+1} = \theta_t - \eta_t g_{\hat{z}}(\theta_t)$ and we can find:

$$
\begin{aligned}
\mathcal{L}_{\hat{z}}(\theta_{t+1}) &\le \mathcal{L}_{\hat{z}}(\theta_t) + \langle\widehat{\nabla}\mathcal{L}_{\hat{z}}(\theta_t), \theta_{t+1} - \theta_t\rangle + \frac{L(l)}{2}\|\theta_t - \theta_{t+1}\|^2 \\
&= \mathcal{L}_{\hat{z}}(\theta_t) - \eta_t\langle\widehat{\nabla}\mathcal{L}_{\hat{z}}(\theta_t), g_{\hat{z}}(\theta_t)\rangle + \frac{(\eta_t)^2 L(l)}{2}\|g_{\hat{z}}(\theta_t)\|^2
\end{aligned}
\tag{22}
$$

where $\eta_t$ represents the learning rate at step $t$. Then, we can take the expectation of Equation 22 for $\hat{z}$ and input $x$:

$$\mathbb{E}_{\hat{z},x}[\mathcal{L}_{\hat{z}}(\theta_{t+1})]$$

$$\leq \mathbb{E}_{\hat{z},x}[\mathcal{L}_{\hat{z}}(\theta_t)] - \eta_t \mathbb{E}_{\hat{z},x}[\|\widehat{\nabla}\mathcal{L}_{\hat{z}}(\theta_t)\|^2] + \frac{(\eta_t)^2 L(l_z)}{2}\mathbb{E}_{\hat{z},x}[\|g_z(\theta_t)\|^2]$$

$$\leq \mathbb{E}_{\hat{z},x}[\mathcal{L}_{\hat{z}}(\theta_t)] - \eta_t \mathbb{E}_{\hat{z},x}[\|\widehat{\nabla}\mathcal{L}_{\hat{z}}(\theta_t)\|^2] + \frac{(\eta_t)^2 L(l)}{2}(4(\hat{d}_t + 4)\mathbb{E}_{\hat{z},x}[\|\widehat{\nabla}\mathcal{L}_{\hat{z}}(\theta_t)\|^2] + 3\epsilon^2 L^2(l)(\hat{d}_t + 4)^3)$$

$$(23)$$

The first inequality is due to the Equation 8 and Equation 22. The second inequality holds because Equation 20 provides the result about $\mathbb{E}_{\hat{z},x}[\|g_z(\theta_t)\|^2]$.

Then, we can select learning rate $\eta_t = \frac{1}{4(\hat{d}_t + 4)L(l)}$ and obtain:

$$\mathbb{E}_{\hat{z},x}[\mathcal{L}_{\hat{z}}(\theta_{t+1})] \leq \mathbb{E}_{\hat{z},x}[\mathcal{L}_{\hat{z}}(\theta_t)] - \frac{1}{8(\hat{d}_t + 4)L(l)}\mathbb{E}_{\hat{z},x}[\|\widehat{\nabla}\mathcal{L}_{\hat{z}}(\theta_t)\|^2] + \frac{3\epsilon^2}{32}L(l)(\hat{d}_t + 4) \quad (24)$$

Then, taking the sum of Equation 24 over the index from $T + 1$ to $0$, we can have that :

$$\mathbb{E}_{\hat{z},x}[\|\widehat{\nabla}\mathcal{L}_{\hat{z}}(\theta_T)\|^2] \leq 8(\hat{d} + 4)L[\frac{\mathcal{L}_{\hat{z}}(\theta_0) - \mathcal{L}_{\hat{z}}^*}{T + 1} + \frac{3\epsilon^2}{32}L(\hat{d} + 4)] \quad (25)$$

where $L(l) \leq L$ for all $\mathcal{L}(\boldsymbol{\theta_t})$. Thus, based on Lemma 2, we can have:

$$\mathbb{E}_{\hat{z},x}[\|\nabla\mathcal{L}_m(\theta_T)\|^2] \leq \frac{\epsilon^2 L^2}{2}(\hat{d} + 4)^3 + 2\mathbb{E}_{\hat{z},x}[\|\widehat{\nabla}\mathcal{L}_{\hat{z}}(\theta_T)\|^2]$$

$$\leq 16(\hat{d} + 4)L\frac{\mathcal{L}_{\hat{z}}(\theta_0) - \mathcal{L}_{\hat{z}}^*}{T + 1} + \frac{\epsilon^2 L^2}{2}(\hat{d} + 4)^2(\hat{d} + \frac{11}{2}) \quad (26)$$

The second inequality is due to the Equation 25. To obtain $\sigma$-accurate solution: $\mathbb{E}_{\hat{z},x}[\|\nabla\mathcal{L}_m(\theta_T)\|^2] \leq \sigma^2$, we can define $\epsilon = \Omega(\frac{\sigma}{\hat{d}^{\frac{3}{2}}L})$.

$$16(\hat{d} + 4)L\frac{\mathcal{L}_{\hat{z}}(\theta_0) - \mathcal{L}_{\hat{z}}^*}{T + 1} + \mathcal{O}(\epsilon^2 L^2 \hat{d}^3) = 16(\hat{d} + 4)L\frac{\mathcal{L}_{\hat{z}}(\theta_0 - \mathcal{L}_{\hat{z}}^*)}{T + 1} + \mathcal{O}(\sigma^2)$$

$$T = \mathcal{O}(\frac{\hat{d}L}{\sigma^2}) \quad (27)$$

Finally, we can finish the proof of the theorem. This theorem illustrates that the presence of pronounced sparsity patterns, along with the smoothness of the objective function, can significantly enhance the rate of convergence, potentially achieving a linear acceleration.

| Model | Method | BoolQ | RTE | WIC | MultiRC | SST-2 | COPA | Average |
|---|---|---|---|---|---|---|---|---|
| LLaMA-7b | Zero-Shot | 65.1 | 49.5 | 50.6 | 55.8 | 79.7 | 59.7 | 60.1 |
| LLaMA-7b | ICL | 67.4 | 54.5 | 52.7 | 58.7 | 81.2 | 84.4 | 66.5 |
| LLaMA-7b | FT | $84.5 \pm 0.0$ | $83.6 \pm 0.9$ | $68.4 \pm 1.3$ | $80.2 \pm 1.4$ | $95.7 \pm 0.3$ | $85.0 \pm 0.8$ | $82.9 \pm 0.8$ |
| LLaMA-7b | MeZO | $75.9 \pm 1.1$ | $71.7 \pm 1.5$ | $61.4 \pm 1.8$ | $69.8 \pm 0.7$ | $94.6 \pm 0.3$ | $86.3 \pm 0.9$ | $76.6 \pm 1.1$ |
| LLaMA-7b | R-MeZO | $76.9 \pm 0.7$ | $75.2 \pm 1.7$ | $62.1 \pm 0.4$ | $68.1 \pm 2.0$ | $94.6 \pm 0.2$ | $84.3 \pm 1.7$ | $76.9 \pm 1.1$ |
| LLaMA-7b | S-MeZO | $\mathbf{80.9 \pm 1.6}$ | $\mathbf{80.7 \pm 1.4}$ | $\mathbf{64.9 \pm 1.5}$ | $\mathbf{73.3 \pm 1.2}$ | $\mathbf{95.0 \pm 0.3}$ | $\mathbf{86.7 \pm 0.7}$ | $\mathbf{80.3 \pm 1.2}$ |

Table 10: Accuracy of Fine-Tuning LLaMA-7b on SuperGLUE (1,000 examples). ICL: In-Context Learning, FT: full-parameter fine-tuning with Adam, R-MeZO: MeZO with Random Mask.

**Algorithm 2** PerturbParameters

**Input:** $\boldsymbol{\theta}$ represents pre-trained LLM weight, perturbation scale $\epsilon$, random seed $s$, mask $\boldsymbol{m}$.
Reset random seed $s$
**for** $\theta_i \in \boldsymbol{\theta}$ **do**
    $z_i \sim \mathcal{N}(0, 1)$
    $\theta_i \leftarrow \theta_i + m_i * \epsilon z_i$
**end for**

**Algorithm 3** GetMask

**Input:** $\boldsymbol{\theta}$ represents pre-trained LLM weight, threshold $\boldsymbol{h}$ ($h_i$ represents threshold of each layer).
**Output:** Mask $\boldsymbol{m}$
**for** $i \leftarrow$ Layer 1 **to** Layer $N$ **do**
    **for** $\theta_{i,j} \in \boldsymbol{\theta_i}$ **do**
        **if** $\theta_{i,j} \leq h_i$ **then**
            $\theta_{i,j} = 1$
        **else**
            $\theta_{i,j} = 0$
        **end if**
    **end for**
**end for**

