# OpenReview forum: "Sparse MeZO: Less Parameters for Better Performance in Zeroth-Order LLM Fine-Tuning"
_ICLR.cc/2025/Conference — Submitted to ICLR 2025_

### Official Review · Reviewer_u9EJ · 2024-10-26

**Soundness:** 2
**Presentation:** 4
**Contribution:** 2
**Rating:** 5
**Confidence:** 4

**Summary:**

The paper builds on the Memory-efficient Zeroth-order optimizer (MeZO) and introduces a new method, called Sparse MeZO. While MeZO is successful in training a neural network only with forward passes and by estimating the batch gradient using a finite difference scheme, there is still a significant gap between MeZO and common first-order methods. The authors aim to narrow this gap by updating only a sparse subset of the model parameters in each iteration. In particular, they propose to only update the smallest weights in each layer. They provide efficient implementations of doing so and show that Sparse MeZO performs better than multiple baselines, including MeZO.

**Strengths:**

The paper is well motivated, the approach is novel and the results are convincing. I particularly enjoyed the writing style and clear structure of the paper, it was mostly easy to follow, enjoyable to read and the reader was not left with many questions. The proposed method is interesting, however there are open questions, which I will discuss below. Overall, I think this is a good paper that however needs to resolve several questions before being suited for publication.

**Weaknesses:**

While I appreciate research in this important field, I have several concerns regarding soundness, clarity and contribution of the work, which I explain in detail below. I hope that these remarks are helpful in improving the work and I am happy to discuss my evaluation.

### Soundness
- In Figure 2b, you compare the potential loss increase on a different batch than the one used by MeZO, you then conclude that "zeroth-order gradient estimates suffer from overfitting or noise" (Line 189). I think to really conclude this, you would need to compare the MeZO gradient on a batch with the actual gradient on the same batch. The very same statement could be true for e.g. SGD, especially if the batch size is small and the variance of the gradient is high. I would greatly appreciate to see this figure instead or in addition to Figure 2b (potentially in appendix).
- In Line 197, you write that "all parameters share the same value". What is meant by this? The formula you say that "is used to estimate the gradient" only gives the magnitude of the multiplier, the actual gradient vector is given by $z$, where definitely not all elements are the same. Apart from that being a false premise, you then conclude that not all parameters are optimized in the true gradient direction, which I found a surprising statement. As far as I understand it, MeZO or variants randomly sample a direction along which the gradient is estimated, constrained to that particular direction. It will with very high probability not be anywhere close to the true gradient direction, by virtue of the fact that you are random sampling the directional derivative.  Please clarify this, maybe I am mistaken here.
- In the same paragraph, you somehow derive that "small weights are less impacted by noise corruption and can generalize better" - it is not clear to my why you derive this conclusion, is it just by the fact that you empirically find this to work better? What is the mathematical intuition or even justification for this statement? I think to resolve this soundness issue, you have to answer to very important questions: First of all, why does it make sense to use less weights in the first place? For First-order Optimization this would be absurd, what is the difference here? Why should this work better than using all weights? And secondly, why would exactly the smallest weights have the highest impact? I could potentially follow this argument if you were actually sparsifying the computed gradient, then one could argue with Taylor approximation or the like, but you are not doing this, you are instead sparsifying the weights. Why?
- Figure 4 highlights the effect of sparsity on the final performance. It is not clear why this curve looks like this and why that would make sense. The authors should clarify why they think that sparsity in the weights during gradient estimation helps (apart from providing empirical evidence that is performs better). What sparsity 70% performs bad, then sparsity 75% performs pretty good, and sparsity 80% performs worse again seems very arbitrary and not intuitive. In the worst case, this is just a noise artifact.


### Clarity and Experimental Validation
- Line 123: What is "SPSA"? Is this defined anywhere in the paper?
- In Line 260, you explain how you determine the mask. Why don't you set the threshold dynamically in each iteration? It's efficient and seems like the obvious thing to do.
- In Table 1, you report values for MeZO-LoRA. While I might have a vague idea of what that could be, this seems to be nowhere explained nor defined. Is this known from the literature or was this just missed?
- Are the numbers in the tables reported with respect to the same overall runtime or number of iterations? If MeZO variants are faster than e.g. FT, it would be nice to see how they compare on the same runtime.
- In the appendix (e.g. F), you refer to Lemma 3.2. What exactly is this, where is this defined? The same holds for the following appendix sections, I think these are referring to theorems/lemmata that do not exist or are not defined with the same numbering style. Please clarify.

### Contribution
- As outlined above, I think the contribution of the work is a major issue here. I fully acknowledge that Sparse MeZO achieves better results and is in that sense a meaningful contribution. However, the derivation of the method seems to be at least somewhat vague, it lacks justification apart from achieving better results. There is not much insight to gain since the paper lacks mathematical justifications, or at least intuitions. The derivation from gradient noise seems to be not very rigorous, at least to me. I hope that the authors can convince me otherwise.


### Minor Remarks
- Section 4.1 uses \citet everywhere where I think \citep is intended, this hinders readability.
- Line 483: There is a typo, I guess it should be "on a single A100 GPU".

**Questions:**

See above.

---

> ### Author Response · Authors · 2024-11-22
> **Response to Reviewer u9EJ**
>
> Thanks for your insightful comments, we carefully address your concerns below.
>
> ### **W1:**
> >In Figure 2b, you compare the potential loss increase on a different batch than the one used by MeZO, you then conclude that "zeroth-order gradient estimates suffer from overfitting or noise" (Line 189). I think to really conclude this, you would need to compare the MeZO gradient on a batch with the actual gradient on the same batch. The very same statement could be true for e.g. SGD, especially if the batch size is small and the variance of the gradient is high. I would greatly appreciate to see this figure instead or in addition to Figure 2b (potentially in appendix).
>
> Thank you for this insightful suggestion. We agree that a more rigorous comparison is needed. Our new analysis compares SGD and MeZO using the same batch, as shown in Figure 5. The results show MeZO has a higher probability of loss increment (0.4-0.6) compared to SGD (0.2-0.3). While both methods can increase loss on different batches, MeZO exhibits this behavior more frequently.
>
> ### **W2:**
> >In Line 197, you write that "all parameters share the same value". What is meant by this? The formula you say that "is used to estimate the gradient" only gives the magnitude of the multiplier, the actual gradient vector is given by z, where definitely not all elements are the same. Apart from that being a false premise, you then conclude that not all parameters are optimized in the true gradient direction, which I found a surprising statement. As far as I understand it, MeZO or variants randomly sample a direction along which the gradient is estimated, constrained to that particular direction. It will with very high probability not be anywhere close to the true gradient direction, by virtue of the fact that you are random sampling the directional derivative. Please clarify this, maybe I am mistaken here.
>
> Thank you for your thoughtful review. Let me clarify the gradient estimation in MeZO.
>
> You are correct that MeZO randomly samples a direction to estimate the gradient. To be more precise, following MeZO's definition and Equation (2), the estimated gradient is: $g = p_g \cdot z$, where $p_g = (L(\theta + \epsilon z) - L(\theta - \epsilon z))/(2\epsilon)$ is a scalar value that's shared across all parameters, while $z$ is the randomly sampled direction vector.
>
> We apologize if our wording was unclear. When we mentioned "shared value", we were referring to the scalar multiplier $p_g$, not the final gradient estimates. Since this scalar multiplier is applied uniformly across dimensions, the resulting update may not align well with the true gradient direction, which could be considered a limitation of the method.

---

> > ### Author Response · Authors · 2024-11-22
> > **Response to Reviewer u9EJ - Continue**
> >
> > ### **W3:**
> > >In the same paragraph, you somehow derive that "small weights are less impacted by noise corruption and can generalize better" - it is not clear to my why you derive this conclusion, is it just by the fact that you empirically find this to work better? What is the mathematical intuition or even justification for this statement? I think to resolve this soundness issue, you have to answer to very important questions: First of all, why does it make sense to use less weights in the first place? For First-order Optimization this would be absurd, what is the difference here? Why should this work better than using all weights? And secondly, why would exactly the smallest weights have the highest impact? I could potentially follow this argument if you were actually sparsifying the computed gradient, then one could argue with Taylor approximation or the like, but you are not doing this, you are instead sparsifying the weights. Why?
> >
> > Thank you for these crucial questions about our proposed approach, below are our responses regarding your concerns:
> >
> > 1. The benefits of using fewer weights in Sparse-MeZO can be understood from both theoretical and intuitive perspectives:
> >
> > - Theoretical Understanding. We provide complete proofs showing that Sparse-MeZO can achieve convergence in $O(dL/\sigma^2)$ steps, where $d$ is the number of parameters in the sub-network (the selected weights with sparse mask). This is a significant theoretical contribution as it demonstrates that focusing on a smaller subset of parameters can accelerate convergence.
> > Based on the above analysis, we can use less weights to reduce the parameter space for gradient estimation, thereby decreasing estimation noise. In addition, related work also illustrates that sparse fine-tuning can achieve comparable performance compared with full fine-tuning [1,2,3]. That motivates us to propose Sparse MeZO to efficiently fine-tune LLM.
> >
> > - Intuitive Understanding. In vanilla MeZO, the gradient estimation $(L(\theta + \epsilon z) - L(\theta - \epsilon z))/(2 \epsilon)$  produces a single scalar value that is applied to all parameters. This leads to two fundamental limitations:
> > First, all parameters share the same coefficient for updates, meaning they are not optimized independently based on their individual importance. More critically, this collective optimization can force some parameters to move in suboptimal or even harmful directions. When a random direction z reduces the overall loss, all parameters are updated along z, even though for some individual parameters $\theta_i$, their corresponding perturbation $z_i$ may actually increase the loss locally.
> >
> > 2. The smallest weights have the highest impact. Our focus on small-magnitude weights is based on their fundamental properties in pre-trained models and zeroth-order optimization:
> >
> > - Large-magnitude weights typically store critical pre-trained information, making them sensitive to perturbations. Even small noise in gradient estimation can significantly disrupt these learned patterns and cause substantial performance drops
> >
> > - In contrast, small-magnitude weights store relatively limited information - even pruning them has minimal impact on model performance. This provides a larger optimization space and more flexibility for updates without risking critical pre-trained knowledge
> >
> > - Small-magnitude weights inherently have larger error tolerance - noise in gradient estimation has proportionally less impact on the model's behavior when applied to smaller weights
> >
> > - As demonstrated in Figure 2c, when continuing training from performance drop points, optimizing small weights leads to better recovery and continued improvement, while updating large weights often results in instability due to disruption of pre-trained patterns.
> >
> > [1] Parameter-Efficient Sparsity for Large Language Models Fine-Tuning,  IJCAI 2022.
> >
> > [2] SparseAdapter: An Easy Approach for Improving the Parameter-Efficiency of Adapters, EMNLP 2022.
> >
> > [3] Sparse is Enough in Fine-tuning Pre-trained Large Language Models,  ICML 2024.

---

> > > ### Author Response · Authors · 2024-11-22
> > > **Response to Reviewer u9EJ - Continue**
> > >
> > > ### **W4:**
> > > >Figure 4 highlights the effect of sparsity on the final performance. It is not clear why this curve looks like this and why that would make sense. The authors should clarify why they think that sparsity in the weights during gradient estimation helps (apart from providing empirical evidence that is performs better). What sparsity 70\% performs bad, then sparsity 75\% performs pretty good, and sparsity 80\% performs worse again seems very arbitrary and not intuitive. In the worst case, this is just a noise artifact.
> > >
> > > Thank you for this important question about the sparsity curve behavior in Figure 4.
> > >
> > > 1.Why they think that sparsity in the weights during gradient estimation helps:
> > >
> > > We provide two key theoretical and practical justifications for why sparsity in weights during gradient estimation helps:
> > >
> > > - Theoretical Foundation: In Appendix F, we provide complete mathematical proofs demonstrating that focusing on a smaller subset of parameters can accelerate convergence. This theoretical analysis shows that sparse updates can lead to more efficient optimization trajectories.
> > >
> > > - Reduced Optimization Interference: In dense gradient estimation, all parameters are updated with the same coefficient, which can be suboptimal. Some parameters may be forced to move in harmful directions due to this collective optimization. Sparsity helps by (1) Reducing the parameter space for gradient estimation, (2) Decreasing estimation noise by focusing on fewer, more relevant parameters, (3) Allowing more independent parameter updates, preventing harmful interference between parameters.
> > >
> > > 2.What sparsity 70\% performs bad, then sparsity 75\% performs pretty good, and sparsity 80\% performs worse again seems very arbitrary and not intuitive:
> > >
> > > We appreciate the reviewer's concern about the performance pattern across different sparsity levels. While 75\% sparsity achieves optimal results, we want to emphasize that other sparsity levels also demonstrate significant and consistent improvements across multiple datasets. Specifically, even the 'suboptimal' sparsity levels of 70\% and 80\% achieve substantial gains over the baseline: approximately 7\% and 10\% performance improvement on RTE, 5\% and 6\% on BoolQ, and 5\% and 6\% on WIC respectively. These consistent improvements across different sparsity levels and diverse datasets strongly suggest that the benefits are systematic rather than noise artifacts.
> > >
> > > The relationship between sparsity and performance can be explained through the balance of two competing factors:
> > >
> > > - At lower sparsity levels (e.g., 70\%), too many parameters are retained during gradient estimation, which can lead to noisy gradient updates due to the interaction between redundant parameters.
> > >
> > > - At higher sparsity levels (e.g., 80\%), too many parameters are pruned, removing important connections needed for effective learning. This over-sparsification limits the model's capacity to capture necessary patterns.
> > >
> > > ### **W5:**
> > > >Line 123: What is "SPSA"? Is this defined anywhere in the paper?
> > >
> > > Thank you for pointing out this lack of clarity. We will add the following definition: SPSA [1] is a classical zeroth-order gradient estimator and also be introduced in the paper of MeZO. It estimates gradients using simultaneous random perturbations for optimization without gradient access:
> > > $g = (L(\theta + \epsilon z) - L(\theta - \epsilon z))/(2\epsilon) \cdot z$.
> > >
> > > [1] Multivariate stochastic approximation using a simultaneous perturbation gradient approximation,  IEEE transactions on automatic control.
> > >
> > > ### **W6:**
> > > >In Line 260, you explain how you determine the mask. Why don't you set the threshold dynamically in each iteration? It's efficient and seems like the obvious thing to do.
> > >
> > > Thank you for this insightful comment about dynamic threshold updates. While dynamic thresholding might seem intuitive, we chose fixed thresholds for several important reasons:
> > >
> > > (1) Hyperparameters: Dynamic threshold usually introduces new hyperparameters and different task or pre-trained models may need different thresholds, which will increase the difficulty of reproducing the results.
> > >
> > > (2) Computational Efficiency: Current approach to determine threshold is one-time cost and we only need to calculate before the starting of training. However, dynamic threshold may introduce additional overhead.
> > >
> > > (3) In this paper, the main aim is to introduce sparse gradient is important for Zeroth-Order optimization based LLM fine-tuning and we would like to propose a method with simple to understand and less hyper-parameters. In the future, we will further explore the benefits of dynamic threshold in MeZO.

---

> > > > ### Author Response · Authors · 2024-11-22
> > > > **Response to Reviewer u9EJ - Continue**
> > > >
> > > > ### **W7:**
> > > > >In Table 1, you report values for MeZO-LoRA. While I might have a vague idea of what that could be, this seems to be nowhere explained nor defined. Is this known from the literature or was this just missed?
> > > >
> > > > We will add the introduction about MeZO-LoRA in the revision. For vanilla LoRA, we can define the weight of each layer as $\theta = \theta_0 + AB$, where A and B are defined in LoRA layer. MeZO-LoRA refers to the combination of MeZO optimization with LoRA as introduced in MeZO paper. This approach applies zeroth-order optimization to the low-rank adaptation matrices instead of the full model parameters, significantly reducing the optimization space while maintaining model performance.
> > > >
> > > > For example, we will sample a random perturbation only on A and B and the perturbed weight can be defined as $\theta = \theta_0 + (A+\epsilon z_A)(B+\epsilon z_B)$, where $z_A$ and $z_B$ are sampled random perturbation on A and B of LoRA.
> > > >
> > > > ### **W8:**
> > > > >Are the numbers in the tables reported with respect to the same overall runtime or number of iterations? If MeZO variants are faster than e.g. FT, it would be nice to see how they compare on the same runtime.
> > > >
> > > > Thank you for this important question about the experimental setup. We want to clarify two key points about our comparison methodology:
> > > >
> > > > - All MeZO variants in our experiments were run with the same number of iterations to ensure fair comparison among zero-order methods. First-order methods (LoRA and FT) were run with fewer iterations, as they typically require fewer steps to converge due to their stronger update directions based on gradients.
> > > >
> > > > - The difference in iteration counts between zero-order and first-order methods reflects an inherent trade-off in our approach. While MeZO achieves significant memory efficiency compared to first-order methods, it requires more iterations to converge due to its use of random direction updates rather than gradient-based updates. This is an expected theoretical property of zero-order optimization methods.
> > > >
> > > > ### **W9:**
> > > > >In the appendix (e.g. F), you refer to Lemma 3.2. What exactly is this, where is this defined? The same holds for the following appendix sections, I think these are referring to theorems/lemmata that do not exist or are not defined with the same numbering style. Please clarify.
> > > >
> > > > We sincerely thank the reviewer for identifying this inconsistency in our cross-referencing. We have now corrected all these references in our revision to ensure consistent numbering throughout the paper.:
> > > > - The reference to "Lemma 3.2" in Appendix F should be "Lemma 1"
> > > > - The reference to "Lemma 3.3" should be "Lemma 2"

---

> > > > > ### Author Response · Authors · 2024-11-22
> > > > > **Response to Reviewer u9EJ - Continue**
> > > > >
> > > > > ### **W10:**
> > > > > >As outlined above, I think the contribution of the work is a major issue here. I fully acknowledge that Sparse MeZO achieves better results and is in that sense a meaningful contribution. However, the derivation of the method seems to be at least somewhat vague, it lacks justification apart from achieving better results. There is not much insight to gain since the paper lacks mathematical justifications, or at least intuitions. The derivation from gradient noise seems to be not very rigorous, at least to me. I hope that the authors can convince me otherwise.
> > > > >
> > > > > We sincerely appreciate the reviewer's thoughtful feedback regarding the theoretical foundations of Sparse MeZO. We acknowledge that our initial presentation could be strengthened, and we will address these concerns in our revision.
> > > > >
> > > > > 1. Mathematical Justification:
> > > > >
> > > > > Our method is built on rigorous mathematical analysis, detailed in Section F of the appendix. We provide complete proofs showing that Sparse-MeZO can achieve convergence in $O(dL/\sigma^2)$ steps, where  $d$ is the number of parameters in the sub-network (the selected weights with sparse mask). This is a significant theoretical contribution as it demonstrates that focusing on a smaller subset of parameters can accelerate convergence.
> > > > >
> > > > > 2. Intuitive Understanding:
> > > > >
> > > > > (1) The Role of Sparse Masking in MeZO:
> > > > >
> > > > > In vanilla MeZO, the gradient estimation $(L(\theta + \epsilon z) - L(\theta - \epsilon z))/(2 \epsilon)$  produces a single scalar value that is applied to all parameters. This leads to two fundamental limitations:
> > > > > First, all parameters share the same coefficient for updates, meaning they are not optimized independently based on their individual importance. More critically, this collective optimization can force some parameters to move in suboptimal or even harmful directions. When a random direction z reduces the overall loss, all parameters are updated along z, even though for some individual parameters $\theta_i$, their corresponding perturbation $z_i$ may actually increase the loss locally.
> > > > >
> > > > > For example, if applying z to all parameters reduces the total loss, MeZO updates every parameter along z. However, this collective decision masks the fact that some parameters might benefit from moving in the opposite direction. Our sparse masking approach addresses these limitations by:
> > > > >
> > > > > - Enabling selective parameter optimization, reducing the interference between parameters
> > > > > - Allowing parameters to be optimized more independently, avoiding forced updates in potentially harmful directions
> > > > > - Permitting larger learning rates for selected parameters without destabilizing training (as shown in Figure 2a).
> > > > >
> > > > > (2) Importance of Small-Magnitude Weights: Our focus on small-magnitude weights is based on their fundamental properties in pre-trained models and zeroth-order optimization:
> > > > >
> > > > > - Large-magnitude weights typically store critical pre-trained information, making them sensitive to perturbations. Even small noise in gradient estimation can significantly disrupt these learned patterns and cause substantial performance drops
> > > > >
> > > > > - In contrast, small-magnitude weights store relatively limited information - even pruning them has minimal impact on model performance. This provides a larger optimization space and more flexibility for updates without risking critical pre-trained knowledge
> > > > >
> > > > > - Small-magnitude weights inherently have larger error tolerance - noise in gradient estimation has proportionally less impact on the model's behavior when applied to smaller weights
> > > > >
> > > > > - As demonstrated in Figure 2c, when continuing training from performance drop points, optimizing small weights leads to better recovery and continued improvement, while updating large weights often results in instability due to disruption of pre-trained patterns.
> > > > >
> > > > > ### **W11:**
> > > > > >-   Section 4.1 uses \citet everywhere where I think \citep is intended, this hinders readability. Line 483: There is a typo, I guess it should be "on a single A100 GPU".
> > > > >
> > > > > Thank you for your careful reading and attention to detail. We have corrected these formatting and typographical issues in the revised version.

---

> > > > > > ### Comment · Reviewer_u9EJ · 2024-11-27
> > > > > >
> > > > > > I thank the authors for their extensive answers. Some remarks:
> > > > > >
> > > > > > > Theoretical Understanding. We provide complete proofs showing that Sparse-MeZO can achieve convergence in [...]
> > > > > >
> > > > > > So what is the tradeoff here? Would you get the best convergence if you update only 1 parameter? Whats the downside?
> > > > > >
> > > > > > > In contrast, small-magnitude weights store relatively limited information - even pruning them has minimal impact on model performance. This provides a larger optimization space and more flexibility for updates without risking critical pre-trained knowledge [... ] Small-magnitude weights inherently have larger error tolerance - noise in gradient estimation has proportionally less impact on the model's behavior when applied to smaller weights
> > > > > >
> > > > > > This is were my question originated from. I agree that by intuition things like magnitude pruning work, since smaller weights have smaller impact on the layer's output than larger weights. How is this related to not updating smaller weights? If you, e.g. prune weights of a pretrained model, these pruned weights will certainly get the highest gradients in the next optimization step. Why do small-magnitude weights have larger error tolerance? I don't see why.
> > > > > >
> > > > > > > W4
> > > > > >
> > > > > > If I am not mistaken, this only partially addresses the issue I raised: "What sparsity 70% performs bad, then sparsity 75% performs pretty good, and sparsity 80% performs worse again seems very arbitrary and not intuitive. In the worst case, this is just a noise artifact. " - how is this explainable?
> > > > > >
> > > > > > > Current approach to determine threshold is one-time cost and we only need to calculate before the starting of training. However, dynamic threshold may introduce additional overhead.
> > > > > >
> > > > > > If you do that every 100 iterations, that is negligible, don't you think? While I agree that having less hyperparameters can be beneficial, I think that should be ablated to see how much you lose by saving hyperparameters.

---

> > > > > > > ### Author Response · Authors · 2024-12-01
> > > > > > > **Response to Reviewer u9EJ**
> > > > > > >
> > > > > > > Thank you very much for your follow-up questions, which have helped us improve our paper.
> > > > > > >
> > > > > > > >1. Theoretical Understanding. We provide complete proofs showing that Sparse-MeZO can achieve convergence in [...]
> > > > > > > So what is the tradeoff here? Would you get the best convergence if you update only 1 parameter? Whats the downside?
> > > > > > >
> > > > > > >
> > > > > > > Thank you for raising this important point.
> > > > > > >
> > > > > > > **So what is the tradeoff here?**:
> > > > > > > - The key trade-off is between sparsity and convergence: too many parameters may allow convergence but slow down optimization, while too few parameters may prevent convergence entirely. This highlights a limitation of our approach - the need to carefully tune the sparsity hyperparameter.
> > > > > > >
> > > > > > > **Would you get the best convergence if you update only 1 parameter? Whats the downside?**:
> > > > > > > - Sparse MeZO's convergence quality depends critically on mask selection. The theoretical analysis **assumes the masked model can converge under first-order optimization**. If first-order optimization cannot make the model converge with a given mask $m$ (e.g., updating only 1 parameter), Sparse MeZO will also fail to converge.
> > > > > > >
> > > > > > > - To validate this relationship, we compared fine-tuning with small-magnitude weights using first-order optimization (Sparse Adam) against full fine-tuning (Adam). The experiments demonstrate that the model achieves comparable performance when using Sparse Adam with 75% sparsity. This suggests that sparse parameter updates can effectively optimize the model while maintaining performance quality.
> > > > > > >
> > > > > > > |Model | Method |   Sparsity   |  RTE  | BoolQ | SST-2 | WIC | Average
> > > > > > > |---------|------|-------|---------|-------|-------|-------|-------------|
> > > > > > > LLaMA-7b | Sparse Adam |  0.75  |  84.0  |  84.5  |  95.3 | 68.2 |  83.0
> > > > > > > LLaMA-7b | Adam |  /  |  83.6  |  84.5  |  95.7  |  68.4  |  83.1  |
> > > > > > >
> > > > > > > >2. In contrast, small-magnitude weights store relatively limited information - even pruning them has minimal impact on model performance. This provides a larger optimization space and more flexibility for updates without risking critical pre-trained knowledge [... ] Small-magnitude weights inherently have larger error tolerance - noise in gradient estimation has proportionally less impact on the model's behavior when applied to smaller weights
> > > > > > > >This is were my question originated from. I agree that by intuition things like magnitude pruning work, since smaller weights have smaller impact on the layer's output than larger weights. How is this related to not updating smaller weights? If you, e.g. prune weights of a pretrained model, these pruned weights will certainly get the highest gradients in the next optimization step. Why do small-magnitude weights have larger error tolerance? I don't see why.
> > > > > > >
> > > > > > > Thank you for your insightful question.
> > > > > > >
> > > > > > > - We think the main reason is that **Zeroth-order gradient estimation introduces more noise compared to exact gradients.** In MeZO, when a random direction z reduces the overall loss, all parameters are updated along z, even though some individual parameters $\theta_i$ may experience locally increased loss from their corresponding perturbation $z_i$. This means certain dimensions of $z_i$ may point toward increasing loss for their respective $\theta_{i}$, introducing additional noise compared to exact gradients. This noisy optimization process risks disrupting valuable pretrained knowledge or fine-tuning progress.
> > > > > > >
> > > > > > > - Our approach targets small weights for updates because they store less critical information: this makes them **safer to modify without disrupting the pretrained knowledge**. Large weights likely encode important patterns we want to preserve, while small weights offer room for optimization with less risk of damaging core model capabilities.  **Based on this intuition, we thnk small-magnitude weights demonstrate greater error tolerance.**
> > > > > > >
> > > > > > > - To empirically validate why updating small-magnitude weights leads to better gradient estimation, we conducted additional experiments. We can extend the MeZO gradient ($g_{mezo}$) as the sum of exact gradient ($g_{true}$) and gradient noise ($\tau$): $g_{mezo} = g_{true} + \tau$. We attribute this primarily to the gradient noise $\tau$. To analyze the effects of $\tau$, we examined the loss change on the validation set at 1000-step intervals: $L_{val}^{(t+1000)} - L_{val}^{(t)}$. A larger magnitude of loss change typically indicates better gradient estimation and error tolerance. The results in the table demonstrate that updating small weights consistently achieves higher loss change magnitude.
> > > > > > >
> > > > > > > - The loss change on validation set (LLaMA-7b, RTE)
> > > > > > >
> > > > > > > | Method |   step=1000   |  step=2000  | step=3000 | step=4000 | step=5000
> > > > > > > |---------|------|-------|---------|-------|-------|
> > > > > > > MeZO | -0.0225  |  -0.0072  |  -0.0068 | -0.0057 |  -0.0034  |
> > > > > > > MeZO (update small weights) |  **-0.0254**  |  **-0.0113**  |  **-0.0069**  |  **-0.0205**  |  **-0.0205**  |

---

> > > > > > > > ### Author Response · Authors · 2024-12-01
> > > > > > > > **Response to Reviewer u9EJ - Continue**
> > > > > > > >
> > > > > > > > > W4
> > > > > > > > >If I am not mistaken, this only partially addresses the issue I raised: "What sparsity 70% performs bad, then sparsity 75% performs pretty good, and sparsity 80% performs worse again seems very arbitrary and not intuitive. In the worst case, this is just a noise artifact. " - how is this explainable?
> > > > > > > >
> > > > > > > > Thank you for your insightful feedback regarding our results. We want to address that our method demonstrates robust performance across multiple sparsity levels. **While 75% sparsity yields optimal results, both 70% and 80% sparsity levels consistently produce substantial improvements across diverse datasets. Specifically, we observe improvements of 7% and 10% on RTE, 5% and 6% on BoolQ, and 5% and 6% on WIC. This range of effective sparsity levels suggests that our method is stable and not overly sensitive to the exact choice of sparsity threshold.**
> > > > > > > >
> > > > > > > > The performance variations across different sparsity levels can be explained by **an inherent trade-off between parameter selection and model performance**. At lower sparsity (70%), the large number of tuned parameters is close to vanilla MeZO's behavior, potentially leading to suboptimal results. Conversely, at higher sparsity (80%), the limited number of updatable parameters may constrain the model's knowledge fitting capacity. This necessitates finding an optimal sparsity level (such as 75%). However, it's worth noting that all tested sparsity levels (70%, 75%, and 80%) consistently show improvements across datasets (7-10% on RTE, 5-6% on BoolQ, and 5-6% on WIC).
> > > > > > > >
> > > > > > > >
> > > > > > > > To address the concern that the improvement is brought by the random seed noise, we expanded our initial single-seed experiments (Figure 4) to include results across three different seeds. These additional experiments across three seeds reveal the model's performance patterns across different sparsity values, with mean accuracies ranging from 79.9% to 80.8% on RTE and 80.9% to 81.4% on BoolQ:
> > > > > > > >
> > > > > > > > - LLaMA on RTE:
> > > > > > > >
> > > > > > > > |Method | seed  | sparsity=0.7 | sparsity=0.75 | sparsity=0.8 |
> > > > > > > > |---------|------|-------|---------|-------|
> > > > > > > > LLaMA-7b + Sparse MeZO | 0 |  79.2  | 83.0 |82.3 |
> > > > > > > > LLaMA-7b + Sparse MeZO | 1 |  81.2  | 80.1 | 80.1 |
> > > > > > > > LLaMA-7b + Sparse MeZO | 2 |  79.4  | 79.1 | 79.9 |
> > > > > > > > Average | / |  79.9 $\pm$ 0.9     |  80.7 $\pm$ 1.6 | 80.8 $\pm$ 1.1
> > > > > > > >
> > > > > > > >
> > > > > > > > - LLaMA on BoolQ:
> > > > > > > >
> > > > > > > > |Method | seed  | sparsity=0.7 | sparsity=0.75 | sparsity=0.8 |
> > > > > > > > |---------|------|-------|---------|-------|
> > > > > > > > LLaMA-7b + Sparse MeZO | 0 |  81.3 | 81.8 | 82.5 |
> > > > > > > > LLaMA-7b + Sparse MeZO | 1 |  80.6  | 81.1 | 79.3 |
> > > > > > > > LLaMA-7b + Sparse MeZO | 2 |  80.8  | 81.3 | 80.8 |
> > > > > > > > Average | / |    80.9 $\pm$ 0.3   | 81.4 $\pm$ 0.3  |  80.9 $\pm$ 1.3 |
> > > > > > > >
> > > > > > > > >Current approach to determine threshold is one-time cost and we only need to calculate before the starting of training. However, dynamic threshold may introduce additional overhead.
> > > > > > > > >If you do that every 100 iterations, that is negligible, don't you think? While I agree that having less hyperparameters can be beneficial, I think that should be ablated to see how much you lose by saving hyperparameters.
> > > > > > > >
> > > > > > > > We agree that calculating the threshold every 100 iterations would introduce negligible computational overhead. To address this point empirically, we analyzed the performance difference between fixed and dynamic thresholds using various update frequencies (k=5, 10, and 100 steps) on RTE. As shown in the table below, all configurations achieve comparable performance, with the fixed threshold showing a slight advantage. We will include additional experimental results in the final version.
> > > > > > > >
> > > > > > > > - RTE:
> > > > > > > >
> > > > > > > > |Method | fixed threshold | k=100 | k=10 | k=5 |
> > > > > > > > |---------|------|-------|---------|-------|
> > > > > > > > LLaMA-7b + Sparse MeZO |  83.0 |  82.0  |  82.0 | 80.2  |

---

### Official Review · Reviewer_VYUc · 2024-11-02

**Soundness:** 2
**Presentation:** 3
**Contribution:** 2
**Rating:** 6
**Confidence:** 4

**Summary:**

This work proposes zeroth order memory-efficient optimization with sparse updates of model parameters for fine-tuning. Sparse updates are shown to facilitate better convergence than vanilla MeZO. The introduced method is evaluated on several tasks from SuperGLUE and several model architectures - Llama-1, Mistral, OPT.

**Strengths:**

* The fact that instability is caused by the noisy optimization of weights with large magnitude appears to be a useful practical insight.

* SparseMeZO consistently improves upon dense MeZO optimization in different setups.

* The proposed memory-efficient implementation of S-MeZO incurs almost zero memory overheads relative to original MeZO and allows tuning large models on a single GPU.

**Weaknesses:**

* The overall contribution is incremental as the work adds additional sparsification steps on top of the MeZO optimization algorithm.

* Evaluation setup is outdated (as for Fall 2024). Llama-1 and Mistral-v0.1 were released more than a year ago, and current models (Llama-3, Llama-3.1, Qwen-2.5, gemma-2) have advanced dramatically in terms of quality since then. In addition, I would suggest testing the approach on a more challenging and representative fine-tuning setup, such as instruction tuning on Alpaca/OASST or any other instruction fine-tuning dataset to fully appreciate the efficacy of SparseMeZO.

* Some important baselines are not taken into consideration. PEFT adapters (LoRA, DoRA) allow for significant memory reduction on optimizer states. Memory footprint on activations and gradients is not very large for relatively short sequences (which is the case for SuperGLUE tasks) and small batches. In addition, it can be further decreased by gradient accumulation at the cost of additional compute. I would suggest adding comparison with these options and reporting memory usage in Table 3.

**Questions:**

* Given a fixed training budget does SparseMeZO outperform fine-tuning with adapters (+ gradient accumulation)?

* I think it would be insightful to add a comparison between the gradient updates with standard SGD optimization and SparseMeZO on a smaller scale model (say of size ~1B parameters).

---

> ### Author Response · Authors · 2024-11-22
> **Response to Reviewer VYUc**
>
> Thanks for your constructive comments, we carefully address your concerns below.
>
> ### **W1:**
> >The overall contribution is incremental as the work adds additional sparsification steps on top of the MeZO optimization algorithm.
>
> We appreciate this feedback and would like to highlight several substantial contributions of our work beyond MeZO:
>
> - Technical Innovation: We identified and addressed a fundamental limitation in ZO optimization - that gradient noise impacts large weights more severely than small weights. This insight led to our selective parameter updating approach.
> - Performance Gains: Our method achieves substantial improvements over MeZO. For example, (1) 9% absolute accuracy improvement on RTE, (2) 3.5x faster convergence, (3) Comparable performance to full fine-tuning while maintaining MeZO's memory efficiency.
> - Novel Implementation: We developed a memory-optimized implementation for sparse masking that maintains inference-level memory consumption, enabling fine-tuning of LLaMA-30b on a single A100 GPU.
> - Theoretical Analysis: We provided rigorous theoretical analysis proving faster convergence rates with sparsification, establishing that our method isn't just an empirical improvement but has sound theoretical foundations.
>
> ### **W2:**
> >Evaluation setup is outdated (as for Fall 2024). Llama-1 and Mistral-v0.1 were released more than a year ago, and current models (Llama-3, Llama-3.1, Qwen-2.5, gemma-2) have advanced dramatically in terms of quality since then. In addition, I would suggest testing the approach on a more challenging and representative fine-tuning setup, such as instruction tuning on Alpaca/OASST or any other instruction fine-tuning dataset to fully appreciate the efficacy of SparseMeZO.
>
> Thank you for this constructive comment. We have extended our evaluation to more recent models, including LLaMA2-7b and LLaMA3-8b. As shown in the table, Sparse-MeZO demonstrates consistent performance improvements on these newer models across RTE, BoolQ, and WIC tasks. For the final version, we will (1) Include comprehensive results on these newer models, (2) Add instruction tuning experiments on Alpaca/OASST datasets, (3) Expand evaluation to additional contemporary models as suggested (Qwen-2.5, Gemma-2).
>
> |Method |   RTE | BoolQ | WIC | Average
> |---------|------|-------|---------|-------|
> LLaMA2 + MeZO | 69.0 | 78.8 | 62.2| 70.0
> LLaMA2 + Sparse MeZO | 77.6 | 82.2 | 65.3| 75.0 ($\uparrow 5.0$)
> LLaMA3 + MeZO | 74.0 | 77.5 | 63.1| 71.5
> LLaMA3 + Sparse MeZO | 81.3 | 82.7 | 65.7| 76.6 ($\uparrow 5.1$)
>
> ### **W3:**
> >Some important baselines are not taken into consideration. PEFT adapters (LoRA, DoRA) allow for significant memory reduction on optimizer states. Memory footprint on activations and gradients is not very large for relatively short sequences (which is the case for SuperGLUE tasks) and small batches. In addition, it can be further decreased by gradient accumulation at the cost of additional compute. I would suggest adding comparison with these options and reporting memory usage in Table 3.
>
> Thank you for these valuable suggestions about additional baselines and memory analysis. We've expanded our evaluation:
>
> - Performance Comparison: Our experiments with LoRA on LLaMA-7b show: While LoRA achieves better performance, Sparse-MeZO significantly narrows the gap between zeroth-order and first-order methods.
>
> |Method | SST-2 | RTE | BoolQ | WIC | MultiRC| Average|
> |---------|------|-------|---------|---------|---------|---------|
> LLaMA + LoRA | 95.0 | 82.3 | 84.5 | 67.6 | 78.3 | 81.5
> LLaMA + MeZO | 94.6 | 71.7 | 75.9 | 61.4 | 69.8 | 74.7
> LLaMA + Sparse MeZO | 95.0  | 80.7 | 80.9 | 64.9 | 73.3 | 79.0
>
> - Memory Analysis: With batch size = 1, Sparse-MeZO reduces memory usage by 25% compared to LoRA. We will include comprehensive memory comparisons with LoRA, DoRA, and gradient accumulation approaches in Table 3 of the final version.
>
> |Method | SST-2 | RTE | BoolQ | WIC | MultiRC| Average|
> |---------|------|-------|---------|---------|---------|---------|
> LLaMA + LoRA | 15.7  | 19.5 | 25.5 | 16.1 | 34.2 | 22.4
> LLaMA + MeZO | 14.6  | 14.6 | 14.6 | 14.6 | 14.6 | 14.6
> LLaMA + Sparse MeZO (EI) | 14.6 | 14.6 | 14.6 | 14.6 | 14.6 | 14.6
>
> ### **Q1:**
> >Given a fixed training budget does SparseMeZO outperform fine-tuning with adapters (+ gradient accumulation)?
>
> Thank you for this insightful comment. While Sparse-MeZO achieves our primary goal of memory efficiency, we acknowledge that LoRA with gradient accumulation would likely achieve faster convergence given the same training budget. Our method's key advantage is enabling fine-tuning of very large models (e.g., LLaMA-30b) on a single GPU where memory constraints are the primary bottleneck, even if this comes at the cost of longer training time. For example, Sparse MeZO can save 25\% memory compared with LoRA (with gradient accumulation).

---

> > ### Author Response · Authors · 2024-11-22
> > **Response to Reviewer VYUc - Continue**
> >
> > ### **Q2:**
> > >I think it would be insightful to add a comparison between the gradient updates with standard SGD optimization and SparseMeZO on a smaller scale model (say of size ~1B parameters).
> >
> > Thank you for this insightful suggestion. The comparative performance of SGD and Sparse-MeZO on LLaMA-7b is shown below:
> >
> > |Method | SST-2 | RTE | BoolQ | WIC | MultiRC| Average|
> > |---------|------|-------|---------|---------|---------|---------|
> > LLaMA + MeZO | 94.6 | 71.7 | 75.9 | 61.4 | 69.8| 74.7
> > LLaMA + Sparse MeZO|95.0 |80.7|80.9| 64.9 | 73.3| 79.0
> > LLaMA + SGD | 95.0 | 83.2 | 83.4 | 66.0  | 75.3| 80.6
> >
> > The results show that Sparse-MeZO significantly outperforms vanilla MeZO and achieves performance close to SGD (79.0 vs 80.6 average accuracy), while maintaining substantially lower memory requirements.

---

> > > ### Comment · Reviewer_VYUc · 2024-11-23
> > > **Response to Rebuttal**
> > >
> > > Thank you for your response. My concerns about the performance of the method and comparison with more baselines have been addressed. Therefore, I have decided to raise the score.

---

> > > > ### Author Response · Authors · 2024-11-26
> > > > **Thank you for raising the score and providing constructive comments!**
> > > >
> > > > Thank you for raising the score and providing constructive comments! We are delighted that our expanded performance evaluation and baseline comparisons have addressed your concerns. If any further questions arise, please feel free to let us know, and we are always happy to discuss them with you.

---

### Official Review · Reviewer_wtUw · 2024-11-03

**Soundness:** 3
**Presentation:** 2
**Contribution:** 2
**Rating:** 6
**Confidence:** 3

**Summary:**

The paper introduces Sparse MeZO, a memory-efficient optimization method for fine-tuning large language models. By selectively applying zeroth-order updates to a carefully chosen subset of parameters, it reduces estimation errors inherent in zeroth-order methods. This approach improves both performance and convergence speed without increasing memory usage, enabling efficient fine-tuning of large models like LLaMA-30b on limited hardware resources.

**Strengths:**

- Interesting Idea: Proposes Sparse MeZO, selectively applying zeroth-order optimization to improve memory efficiency.-
- Good Performance: Achieves significant gains in accuracy and convergence speed over existing methods.
- Simple but Efficient Algorithm: Offers a straightforward yet effective approach for fine-tuning large models on limited hardware.

**Weaknesses:**

Some details need to be clarify, as explained in Questions

**Questions:**

- Mask Updates per Iteration: Are the masks updated in each iteration? The concept of dynamic masking appears similar to Dynamic Sparse Training (DST), where the frequency of mask updates is a critical factor affecting performance [1,2]

- Can Sparse MeZO be integrated with LoRA, similar to how MeZO has been combined with LoRA in previous work?

-  When claiming the 3.5× speed-up in Figure 1, are both methods using the same learning rate, or does the speed-up depend on differing learning rates?

-  Determining the masks is a key component of this work, and the authors use a simple magnitude-based method. Do you believe that employing more advanced methods like Wanda [3] or SparseGPT [4] could further enhance performance?

[1] Do we actually need dense over-parameterization? in-time over-parameterization in sparse training, ICML 2021

[2] Rigging the Lottery: Making All Tickets Winners， ICML 2020

[3] A Simple and Effective Pruning Approach for Large Language Models, ICLR2024

[4] SparseGPT: Massive Language Models Can Be Accurately Pruned in One-Shot

---

> ### Author Response · Authors · 2024-11-22
> **Response to Reviewer wtUw**
>
> Thanks for your constructive feedback, we carefully address your concerns below.
>
> ### **Q1:**
> >Mask Updates per Iteration: Are the masks updated in each iteration? The concept of dynamic masking appears similar to Dynamic Sparse Training (DST), where the frequency of mask updates is a critical factor affecting performance [1,2]
>
> Thanks for your insightful comment. In our implementation, while the sparsity threshold remains fixed, the mask updates dynamically each iteration using: `mask = (torch.abs(weight) < threshold).float()`.
>
> We appreciate the connection to Dynamic Sparse Training (DST) [1,2] and provide the discussion in the revision. While DST's findings about update frequency are valuable, we chose an update frequency of 1 to maintain memory efficiency. This allows us to compute masks on-the-fly without storing them, using only the model weights and threshold values. Our experiments comparing update frequencies on LLaMA-7b show that frequency=1 outperforms frequency=5, as shown in the table. We will incorporate a discussion of DST and mask update frequency in our revision.
>
> |Method |  SST-2 | RTE | BoolQ | WIC | MultiRC | Average|
> |---------|------|-------|---------|-------|-------|-------|
> Frequency = 1 | 95.0 | 80.7 | 80.9 | 64.9 | 73.3 | 79.0
> Frequency = 5 | 95.0 |  76.1| 77.1 | 62.7 | 70.8 | 76.3
>
> ### **Q2:**
> >Can Sparse MeZO be integrated with LoRA, similar to how MeZO has been combined with LoRA in previous work?
>
> Thanks for your constructive comment. Sparse-MeZO can indeed be integrated with LoRA by applying the masking mechanism to the weight matrices: $\theta = \theta_0 + m \odot (AB)$. We also conduct experiments to compare these methods:
>
> |Method |  SST-2 | RTE | BoolQ | WIC | MultiRC | Average|
> |---------|------|-------|---------|-------|-------|-------|
> MeZO + LoRA | 95.0 | 74.9 | 77.9 | 60.8 | 72.6 | 76.2
> Sparse MeZO + LoRA | 95.0 | 78.2 | 81.3 | 63.3 | 73.1 | 78.2
> Sparse MeZO | 95.0 | 80.7 | 80.9 | 64.9 | 73.3 | 79.0
>
> Our experimental results show that Sparse-MeZO with LoRA achieves comparable performance to vanilla Sparse-MeZO while outperforming MeZO+LoRA.
>
> ### **Q3:**
> >When claiming the 3.5× speed-up in Figure 1, are both methods using the same learning rate, or does the speed-up depend on differing learning rates?
>
> We use a larger learning rate for sparse-mezo. For example, sparse mezo can use a 5x learning rate when only update 20\% or 25\% parameters on RTE task. As demonstrated in Figure 2(a), vanilla MeZO diverges with larger learning rates, while Sparse-MeZO remains stable. This ability to use larger learning rates is a key factor enabling the 3.5x speedup.
>
> ### **Q4:**
> >Determining the masks is a key component of this work, and the authors use a simple magnitude-based method. Do you believe that employing more advanced methods like Wanda [3] or SparseGPT [4] could further enhance performance?
>
> Thank you for this insightful comment about advanced pruning methods. We also provide the discussion in the revision.
>
> -   Our experimental results demonstrate that updating large-magnitude weights, which is a key component of many advanced pruning methods, actually leads to worse performance in zeroth-order optimization compared to focusing on small-magnitude weights.
> -   While Wanda and SparseGPT offer sophisticated pruning strategies, implementing their scoring mechanisms during training would add computational overhead compared to our simple magnitude-based threshold approach.
> -   Our method achieves strong performance while maintaining memory efficiency, suggesting that more complex pruning strategies may not be necessary for effective zeroth-order optimization.

---

> > ### Comment · Reviewer_wtUw · 2024-11-27
> >
> > Thanks for your response.
> > Most of my concerns have been addressed. I would like to increase my score. I suggest including a discussion of Wanda of SparseGPT in the manuscript.

---

### Official Review · Reviewer_ZUwR · 2024-11-08

**Soundness:** 3
**Presentation:** 4
**Contribution:** 2
**Rating:** 5
**Confidence:** 3

**Summary:**

The paper proposes Sparse-MeZO, a memory-efficient zeroth-order optimization (ZO) technique for fine-tuning LLM by selectively optimizing a subset of parameters, rather than all. Based on MeZO, Sparse-MeZO introduces a sparse mask that targets smaller, noise-resilient weights, thereby mitigating gradient estimation noise, improving convergence speed, and reducing memory usage. Key results demonstrate Sparse-MeZO’s efficiency, achieving faster convergence and improved performance over MeZO, with the ability to fine-tune models like LLaMA-30b on limited hardware resources (e.g., a single A100 GPU).

**Strengths:**

1. Incorporating sparsity and MeZO is an interesting direction for performance improvement.

2. The paper offers good empirical validation, showing clear improvements over baseline methods across a range of fine-tuning tasks.

3. The paper is structured logically, making it easy to follow the motivation and methodology

**Weaknesses:**

1. The paper lacks details on how thresholds for selecting small-magnitude weights are determined, how they vary across tasks, and whether they require re-tuning for different settings.

2. It’s not clear if masking and selecting small-magnitude weights specifically benefit zeroth-order optimization more than generic fine-tuning (first-order methods). Since subset selection often improves generalization, it would be needed to evaluate this effect.

3. There is no specific numbers regarding the computational overhead introduced by dynamic masking and threshold calculation.

4. The motivation for why zeroth-order optimization particularly benefits from small-magnitude weights lacks enough theoretical or empirical support. An ablation study showing this effect more clearly would strengthen the argument.

**Questions:**

1. Could the authors clarify the computational cost associated with generating the dynamic mask at each iteration and how it compares to MeZO in practice? such as a wall-clock time wise comparison similar to Figure 1

2. How sensitive is Sparse-MeZO’s performance to the choice of layer-wise sparsity threshold, and what considerations guided the threshold selection?

3. Can you provide detailed threshold hyperparameters for reproducibility?

---

> ### Author Response · Authors · 2024-11-22
> **Response to Reviewer ZUwR**
>
> Thanks for your constructive feedback, we carefully address your concerns below.
>
>
> ### **W1 & Q2:**
> > The paper lacks details on how thresholds for selecting small-magnitude weights are determined, how they vary across tasks, and whether they require re-tuning for different settings.
> > How sensitive is Sparse-MeZO’s performance to the choice of layer-wise sparsity threshold, and what considerations guided the threshold selection?
>
> Thank you very much for your valuable comment. We determine thresholds using a principled sparsity-based approach. Specifically, we use a percentile-based method where the threshold is set based on a target sparsity level. For example, with 80\% sparsity, we sort the weight values of each layer and set the threshold at the 80th percentile. Importantly, this threshold is determined once before training begins and remains fixed throughout the optimization process.
>
> Our empirical results in Figure 4 demonstrate that a sparsity level of 75\% consistently yields strong performance across diverse tasks (RTE, BoolQ, and WIC), highlighting the generalizability of our threshold selection method. Furthermore, we observe robust performance gains compared to vanilla MeZO across a range of sparsity values (70-80\%), indicating that our method is not sensitive to the exact threshold choice. This stability across different tasks eliminates the need for task-specific threshold tuning.
>
> To further illustrate with a concrete example: if we have a layer with 1000 weights and set 75\% sparsity, we would sort all weight magnitudes and set the threshold at the 750th value (75th percentile). Any weight with magnitude below this threshold is considered small. This percentile-based approach automatically adapts to each layer's weight distribution, eliminating the need for manual tuning across different tasks or model sizes. As demonstrated in Figure 4, this single approach with 75\% sparsity works robustly across RTE, BoolQ, and WIC tasks, yielding consistent improvements over vanilla MeZO.
>
> |Sparsity  |SST-2  |RTE | BoolQ   | WIC |
> |---------|------|-------|---------|-------|
> |0.6  | 95.0 ($\uparrow$ 0.4) | 79.1 ($\uparrow$ 7.4)| 80.1 ($\uparrow$ 4.2) | 63.7 ($\uparrow$ 2.3) |
> |0.7  | 95.3 ($\uparrow$ 0.7) | 79.4 ($\uparrow$ 7.7)| 81.3 ($\uparrow$ 5.4) | 63.5 ($\uparrow$ 2.1) |
> |0.75 | 95.2 ($\uparrow$ 0.6) | 83.0 ($\uparrow$ 11.3) | 81.7 ($\uparrow$ 5.8)| 66.0 ($\uparrow$ 4.6) |
> |0.8  | 95.0 ($\uparrow$ 0.4) | 82.3 ($\uparrow$ 10.6) | 82.5 ($\uparrow$ 6.6)| 64.1 ($\uparrow$ 2.7) |
> |MeZO | 94.6       | 71.7        | 75.9      | 61.4       |
>
> ### **W2:**
> >It’s not clear if masking and selecting small-magnitude weights specifically benefit zeroth-order optimization more than generic fine-tuning (first-order methods). Since subset selection often improves generalization, it would be needed to evaluate this effect.
>
> Thank you for this comment about distinguishing the specific benefits of small-magnitude weight selection across optimization methods. Our additional experiments compare this selection strategy's impact on both zeroth-order (MeZO) and first-order (Adam) optimization, with results shown in the table. While weight selection provides some generalization benefits for first-order methods, our findings demonstrate that the improvements are markedly larger for zeroth-order optimization, suggesting this approach particularly addresses zeroth-order optimization challenges.
>
> |Method   |Sparsity |LR    | SST-2   | RTE  | BoolQ | WIC | MultiRC | Average |
> |---------|------|-------|---------|-------|-------|-------|-------|-------|
> MeZO | / | / | 94.6 | 71.7 | 75.9 | 61.4 | 69.8 | 74.7
> Sparse MeZO | / | / | 95.0 | 80.7 | 80.9 | 64.9 | 73.3 | 79.0
> Adam        | /|1e-4| 95.7 | 83.6 | 84.5 | 68.4 | 80.2 | 82.5
> Sparse Adam |0.7|1e-4| 95.6| 81.6 | 84.5 | 64.9 | 77.7 | 80.9 ($\downarrow1.6$)
> Sparse Adam |0.7|2e-4| 95.6| 85.1 | 85.1 | 66.9 | 80.7 | 82.6
> Sparse Adam |0.8|1e-4| 95.6| 81.6 | 83.9 | 62.3 | 77.2 | 80.1 ($\downarrow2.4$)
> Sparse Adam |0.8|2e-4| 95.6| 84.0 | 84.7 | 68.1 | 81.2 | 82.7
>
> The results in the table demonstrate three key findings:
>
> 1. For zeroth-order optimization, Sparse MeZO improves MeZO's average performance from 74.7 to 79.0, showing clear benefits of weight masking.
> 2. For first-order optimization with Adam using a fixed learning rate (1e-4), masking actually reduces performance. With 0.7 sparsity, average performance drops from 82.5 to 80.9, and with 0.8 sparsity, it further decreases to 80.1.
> 3. While increasing the learning rate (from 1e-4 to 2e-4) with masking in first-order optimization does improve results (from 80.9 to 82.6 at 0.7 sparsity, and 80.1 to 82.7 at 0.8 sparsity), the gains are minimal (0.1-0.2 improvement). This confirms that masking provides significantly greater benefits for zeroth-order optimization.

---

> > ### Author Response · Authors · 2024-11-22
> > **Response to Reviewer ZUwR - Continue**
> >
> > ### **W3 & Q1:**
> > >There is no specific numbers regarding the computational overhead introduced by dynamic masking and threshold calculation.
> > >Could the authors clarify the computational cost associated with generating the dynamic mask at each iteration and how it compares to MeZO in practice? such as a wall-clock time wise comparison similar to Figure 1
> >
> > Thank you for your constructive comment about computational overhead. We have conducted detailed measurements to quantify the overhead and the results is shown in the folowing table. Due to the main overhead is from the definition of threshold and mask calculation, we mainly analyze them in the table:
> >
> > |Method|  SST-2 | RTE | BoolQ | WIC | MultiRC |
> > |---------|------|-------|---------|-------|-------|
> > LLaMA + MeZO-Threshold | 0.0139s | 0.0141s | 0.0142s | 0.0135s | 0.0143s
> > LLaMA + MeZO-Mask (each step) | 0.0714s | 0.0673s | 0.0701s | 0.0717s | 0.0643s
> > LLaMA + MeZO (each step) | 0.8588s | 2.3631s | 3.6336s | 1.0899s | 6.2621s
> > Overhead = t(Mask)/t(MeZO) | 0.08 | 0.025 | 0.019 | 0.06 | 0.009
> >
> > (1) Threshold Calculation (One-time Cost):
> > The threshold determination process takes approximately 0.01 seconds for a LLaMA-7b model. In addition, it is a one-time cost before training begins, and therefore this additional computation is very limited compared with several hours training of each experiment.
> >
> > (2) Dynamic Masking Overhead:
> > We provide the detailed dynamic masking calculation time and each training step time in the table. From this table, we can find that the masking operation adds approximately 0.07s per forward pass, and the training step time in these experiments is dynamic from 0.8588s to 6.2621s. Therefore, the relative overhead is dynamic from 0.009 to 0.08.
> >
> > ### **W4:**
> > >The motivation for why zeroth-order optimization particularly benefits from small-magnitude weights lacks enough theoretical or empirical support. An ablation study showing this effect more clearly would strengthen the argument.
> >
> > Thank you very much for this valuable suggestion.
> >
> > 1. Our focus on small-magnitude weights is based on their fundamental properties in pre-trained models and zeroth-order optimization:
> >
> > - Large-magnitude weights typically store critical pre-trained information, making them sensitive to perturbations. Even small noise in gradient estimation can significantly disrupt these learned patterns and cause substantial performance drops
> >
> > - In contrast, small-magnitude weights store relatively limited information - even pruning them has minimal impact on model performance. This provides a larger optimization space and more flexibility for updates without risking critical pre-trained knowledge
> >
> > - Small-magnitude weights inherently have larger error tolerance - noise in gradient estimation has proportionally less impact on the model's behavior when applied to smaller weights
> >
> > - As demonstrated in Figure 2c, when continuing training from performance drop points, optimizing small weights leads to better recovery and continued improvement, while updating large weights often results in instability due to disruption of pre-trained patterns.
> >
> > 2. Ablation Study: Our experimental analysis examines how weights of different magnitudes affect performance. We divided weights into 4 groups by magnitude, where '0-0.25' represents the bottom 25% and '0.75-1.0' represents the top 25%. The results in the tables demonstrate that updating small-magnitude weights consistently achieves better performance than updating large-magnitude weights.
> >
> > || 0-0.25 | 0.25-0.5 | 0.5-0.75 | 0.75-1.0 |
> > |---------|------|-------|---------|-------|
> > RTE |  82.7  | 73.6 | 66.1  | 57.3   |
> > BoolQ |81.6 | 77.1 | 67.7  |  66.1   |
> >
> > Building on these findings, we performed additional experiments to analyze weight selection strategies. While updating small-magnitude weights (0-0.25) shows strong performance, we tested incrementally including more weight groups (e.g., expanding to 0-0.5, which includes both 0-0.25 and 0.25-0.5). The results show that adding weights beyond the small-magnitude group (0-0.25) actually reduces performance, confirming that focusing on small-magnitude weights is optimal.
> >
> > || 0-0.25 | 0-0.5 | 0-0.75 | 0-1.0 |
> > |---------|------|-------|---------|-------|
> > RTE |  82.7  | 75.1 | 72.3  |  71.7  |
> > BoolQ |81.6 | 77.8 |  76.7 |   75.9  |

---

> > > ### Author Response · Authors · 2024-11-22
> > > **Response to Reviewer ZUwR - Continue**
> > >
> > > ### **Q3:**
> > > >Can you provide detailed threshold hyperparameters for reproducibility?
> > >
> > > Thanks for your constructive suggestion. We provide complete threshold-related hyperparameters (based on sparsity value) in Table 8 of the Appendix:
> > >
> > > |Method | SST-2 | RTE | BoolQ | WIC |
> > > |---------|------|-------|---------|-------|
> > > LLaMA + Sparse MeZO | 0.70   | 0.75 |  0.80 |   0.80 | 0.80 |
> > > Mistral + Sparse MeZO | 0.70 | 0.60 |  0.60 |   0.70 | 0.60 |
> > >
> > > From the table, we can find the best sparsity value for LLaMA-7b is between 0.70 and 0.80, and the best sparsity value for Mistral is between 0.60 and 0.70.

---

### Meta-Review · Area_Chair_ksoR · 2024-12-20

**Metareview:**

This paper proposes Sparse-MeZO, a variant of a recent zeroth-order optimizer (MeZO) used in fine tuning LLMs. Different from MeZO, the authors' method involves only performing zeroth order optimization over a subset of the learnable parameters. Since reducing parameter dimensionality is dramatically beneficial to all zeroth order optimization routines, this results in faster convergence and superior performance. Strengths obviously include the good empirical evaluation with clear improvements over MeZO. While I think this paper is well on its way to acceptance given the strong empirical results, there were significant concerns about (1) the clarity of certain key details that I think *still* remain unsatisfactorily addressed even in the updated paper (see below), and (2) key details above the algorithm.

**Additional Comments On Reviewer Discussion:**

Based on the reviewer concerns and author feedback period, I would suggest the following changes to significantly strengthen the paper:

(1) Include the results from the author feedback on LLaMa 2 and 3, as well as the comparison to LoRA. I think these results substantially update and strengthen the empirical evaluation.

(2) The updated discussion in the paper on threshold selection is still too vague: (2a) the word "principled" has no scientific content, (2b) saying "we use a percentile-based method" does not describe the method, nor how the percentile was picked, nor why the particular desired sparsity level was chosen. Was it the maximum density that will fit on a single A100? I assume not, because of the *optimization* benefits of additional sparsity.

(3) Some claims require better evidence. The specific example I am thinking of here is the point made by u9EJ, that it would be better to compare MeZO estimated gradients to actual gradients on the same minibatch.

(4) Related to (2), provide an explanation in the paper for the nonmonotonic impact of sparsity level (Figure 4) on performance. This nonmonotonicity presumably makes it more challenging to pick the threshold as discussed in point (2).

---

### Decision · Program_Chairs · 2025-01-22

Reject